# Optimized design and in vivo application of optogenetically functionalized *Drosophila* dopamine receptors

Fangmin Zhou [1,2,3], Alexandra-Madelaine Tichy[4,5], Bibi Nusreen Imambocus[2], Shreyas Sakharwade[1,2], Francisco J. Rodriguez Jimenez[6,7], Marco González Martínez[6], Ishrat Jahan[6], Margarita Habib[8], Nina Wilhelmy[9], Vanessa Burre[9], Tatjana Lömker [3], Kathrin Sauter[3], Charlotte Helfrich-Förster [8], Jan Pielage [9], Ilona C. Grunwald Kadow [6,7], Harald Janovjak[4,5,10] & Peter Soba [1,2,3] ✉

Neuromodulatory signaling *via* G protein-coupled receptors (GPCRs) plays a pivotal role in regulating neural network function and animal behavior. The recent development of optogenetic tools to induce G protein-mediated signaling provides the promise of acute and cell type-specific manipulation of neuromodulatory signals. However, designing and deploying optogenetically functionalized GPCRs (optoXRs) with accurate specificity and activity to mimic endogenous signaling in vivo remains challenging. Here we optimize the design of optoXRs by considering evolutionary conserved GPCR-G protein interactions and demonstrate the feasibility of this approach using two *Drosophila* Dopamine receptors (optoDopRs). These optoDopRs exhibit high signaling specificity and light sensitivity in vitro. In vivo, we show receptor and cell type-specific effects of dopaminergic signaling in various behaviors, including the ability of optoDopRs to rescue the loss of the endogenous receptors. This work demonstrates that optoXRs can enable optical control of neuromodulatory receptor-specific signaling in functional and behavioral studies.

Behavioral flexibility, learning, as well as goal-directed and state-dependent behavior in animals depend to a large degree on neuromodulatory signaling via G protein-coupled receptors (GPCRs), which tune neuronal network function to the current external sensory environment and the internal state of the animal[1]. Dopamine (DA) is one of the most conserved metabotropic neurotransmitters and modulators, which can activate different G protein-dependent and -independent signaling events via its cognate GPCRs[2,3]. Depending on

[1]Institute of Physiology and Pathophysiology, Friedrich-Alexander-Universität Erlangen-Nürnberg, 91054 Erlangen, Germany. [2]LIMES Institute, Department of Molecular Brain Physiology and Behavior, University of Bonn, Carl-Troll-Str. 31, 53115 Bonn, Germany. [3]Neuronal Patterning and Connectivity laboratory, Center for Molecular Neurobiology (ZMNH), University Medical Center Hamburg-Eppendorf, 20251 Hamburg, Germany. [4]Australian Regenerative Medicine Institute (ARMI), Faculty of Medicine, Nursing and Health Sciences, Monash University, 3800 Clayton, Victoria, Australia. [5]European Molecular Biology Laboratory Australia (EMBL Australia), Monash University, 3800 Clayton, Victoria, Australia. [6]Institute of Physiology II, University Clinic Bonn (UKB), University of Bonn, 53115 Bonn, Germany. [7]ZIEL-Institute of Life and Health, Technical University of Munich, School of Life Sciences, 85354 Freising, Germany. [8]Neurobiology and Genetics, Biocenter, University of Würzburg, Am Hubland, 97074 Würzburg, Germany. [9]Division of Neurobiology and Zoology, RPTU University of Kaiserslautern, 67663 Kaiserslautern, Germany. [10]Flinders Health and Medical Research Institute, College of Medicine and Public Health, Flinders University, 5042 Bedford Park, South Australia, Australia. ✉e-mail: peter.soba@fau.de

the receptor subtype, DA signaling can thereby increase or decrease the excitability of the affected neuronal substrates as well as induce synaptic plasticity and long-term transcriptional changes. Typically, activation of D1-like receptors leads to an increase in cyclic adenosine monophosphate (cAMP) levels through activation of adenylate cyclase (AC), while D2-like receptors inhibit AC and thus decrease cAMP levels[2]. Thereby, DA regulates numerous functional processes, including motivation, locomotion, learning and memory via its distinct cognate receptors[2–6]. Dysregulated DA signaling has been linked to several neurological conditions, including schizophrenia, ADHD, and Parkinson's disease[2]. Due to the differential expression and signaling properties of DA receptors affecting distinct circuits and behaviors, systemic DA pathway modulation can result in unwanted and unspecific side effects. Thus, it is highly desirable to obtain more precise insight into the action of DA signaling and that of other neuromodulators on a receptor-specific basis. However, pharmacological approaches are not cell type-specific and difficult to control temporally, thus lacking the precision and specificity to target defined circuits and their regulated behaviors. At the same time, most current genetic tools do not offer the temporal control and sensitivity required to manipulate the corresponding receptors directly and acutely with high efficiency in vivo.

Optogenetics has revolutionized our understanding of the function of specific neural circuits, allowing for investigation of their role in behavior and physiology through genetic targeting and high spatiotemporal precision[7–9]. While cell type-specific manipulation of neurons in vivo using light-controlled ion channels has evolved rapidly, and numerous powerful tools are available, optical control of modulatory GPCR mediated signaling in general, and in circuits endogenous to the modulatory neurotransmitter, has been more limited so far[10–12]. This is in part due to the difficulty of designing functional light-activatable GPCRs showing endogenous-like localization and activity of the target receptor. Previous studies established chimeric receptor designs in which the intracellular domains of a receptor of interest were swapped into a prototypical light-sensitive GPCR, typically bovine Rhodopsin (Rho). In one example, this strategy has been successfully applied to the β2-adrenergic receptor (β2AR) and has yielded a functional optoXR displaying signaling comparable to its native counterpart[13–17]. A systematic approach for class A GPCRs has produced a library of human optoXRs displaying in vitro signaling capacity corresponding to orphan receptors[18]. Similarly, functional class A/F chimera (Rho:-Frizzled7) and class A/C chimera (Opn4:mGluR6) were designed and applied in optogenetic cellular migration and vision restoration studies, respectively[19,20]. Additional approaches have used structure-guided design, primary sequence-based empirical methods or native light-sensitive GPCRs with similar signaling properties as the receptor of interest[10,11,17]. While it is appealing to utilize optoXRs to mimic GPCR function, design and functionality remain challenging. Importantly, the signaling properties of many GPCRs depend on the cell type, receptor localization and activation kinetics as well as the functional context[11,21–24]. Only in a few cases have optoXRs been deployed in vivo, and they have so far mostly been used to manipulate G protein signaling pathways without perturbation of the endogenous receptor signaling (see Supplementary Table 1). Thus, there is very limited evidence that optoXRs can functionally replace or mimic endogenous GPCR function in target tissues.

In vivo models, including *Drosophila melanogaster*, have contributed extensively to our understanding of neuromodulatory GPCR signaling in neural circuit function and behavior[1,25–29]. In particular, DA and its receptors have been long studied in *Drosophila* regarding their role in learning, memory and goal-directed behaviors[3,5,6,30–33]. *Drosophila* encodes 4 Dopamine receptors: two D1-like receptors (Dop1R1 and Dop1R2), a D2-like receptor (Dop2R) and Dopamine-Ecdysteroid receptor (DopEcR). Dop1R1 and Dop1R2 display conserved functions in learning and memory in the insect learning center, the mushroom body (MB), by inducing cAMP and intracellular calcium store release, respectively[31,34–40]. Dop1R1 is particularly important for the acquisition of new memories[34], while Dop1R2 is involved in transient and permanent forgetting of learned associations in flies[34,39,41]. In addition, both receptors play opposing roles in directing synaptic and behavioral plasticity in the MB during olfactory association[37], and Dop1R1 has also been implicated in larval locomotion[42]. Yet so far, most acute (i.e., dynamic and short-term) cell type-specific functions of these receptors, such as the timing and duration of their signaling, could not be manipulated due to the lack of suitable tools. OptoXRs that can be readily expressed in vivo and allow precise spatiotemporal dissection of endogenous-like dopaminergic signaling and function would solve these issues but are currently not available.

Here, we generate and optimize chimeric optoXRs of *Drosophila melanogaster* Dop1R1 and Dop1R2 by taking advantage of evolutionary constraints of G protein-coupling specificity. We characterize opto-DopR signaling in vitro and find that our optimized design results in improved signaling specificity and light-dependent G protein activation. In vivo, expression and subcellular localization to axonal and dendritic compartments were strongly improved, more closely resembling the endogenous receptor distribution. We then demonstrate that optoDopRs in vivo can replace or mimic dopamine receptor functionality in various DA-dependent behaviors, including locomotion, arousal, learning and operant feeding behavior. Intriguingly, we find cell type and receptor-specific functions using our optoDopRs in innate and adaptive behaviors showing their utility to study DA-dependent function and behavior with high spatiotemporal precision and specificity.

## Results

### Optimization of sequence-based design for optoDopRs

Previous studies have developed sequence-[14,20,43] or structure-based[17] rules for exchanging regions of GPCRs to generate various chimera that display functional signaling of the target receptor yet altered ligand/sensor specificity. Most optoXRs developed so far were built on Rho as a light-sensitive backbone, mainly due to its well-described structure and function, together with sequence-based rules developed by Kim et al.[14,16,18,44]. In the original design rules, transmembrane (TM) helices and intracellular loop (ICL) regions were exchanged. This resulted in chimeric receptors in which at least two or all three ICLs with proximal TM residues and the C-terminus of Rho were substituted by the corresponding regions of the target receptor. We applied this methodology (termed here 'V1') to *Drosophila* Dop1R1 (Fig. 1a) and Dop1R2 as well as six further *Drosophila* GPCRs (AkhR, 5-HT1B, Lgr3, Lgr4, sNPFR, and TkR99D) and generated corresponding optoXR chimera. To test their functionality in cells, we utilized chimeric $G_{\alpha s}$ proteins ('$G_{sx}$ assay') consisting of the signaling domain of $G_s$ fused to the GPCR binding sequence of a specific $G_\alpha$ protein (s/i/t/o/z/q/12/13/15), thus redirecting all signaling toward cAMP increase (Fig. 1b)[45]. Co-expression of $G_{sx}$ chimera with the GPCR of interest in HEK293T or G protein-deficient cells (HEK293ΔG7)[46] for $G_s$-coupled receptors thus allows direct comparison of coupling specificity and strength using the cAMP reporter GloSensor[43]. Except for optoDop1R1[V1], we failed to detect any major G protein signaling in all other optoXRs, (Fig. 1c, Supplementary Fig. 1a–g). Therefore, we revised the receptor design based on recently computed evolutionary constraints of G protein binding to receptors[47]. It became evident that ICL1 was generally not contributing to major G protein binding contacts, so we reasoned that retaining Rho ICL1 should not limit signaling but may increase the structural integrity of a chimeric optoXR. In addition, we readjusted the TM7/C-terminus exchange site to accommodate additional G-protein contact sites. These sites have been defined in the evolutionary analysis of GPCR-G protein interactions through inspection of multiple GPCR-G-protein complex structures of class A receptors. Using this approach (termed 'V2'), we

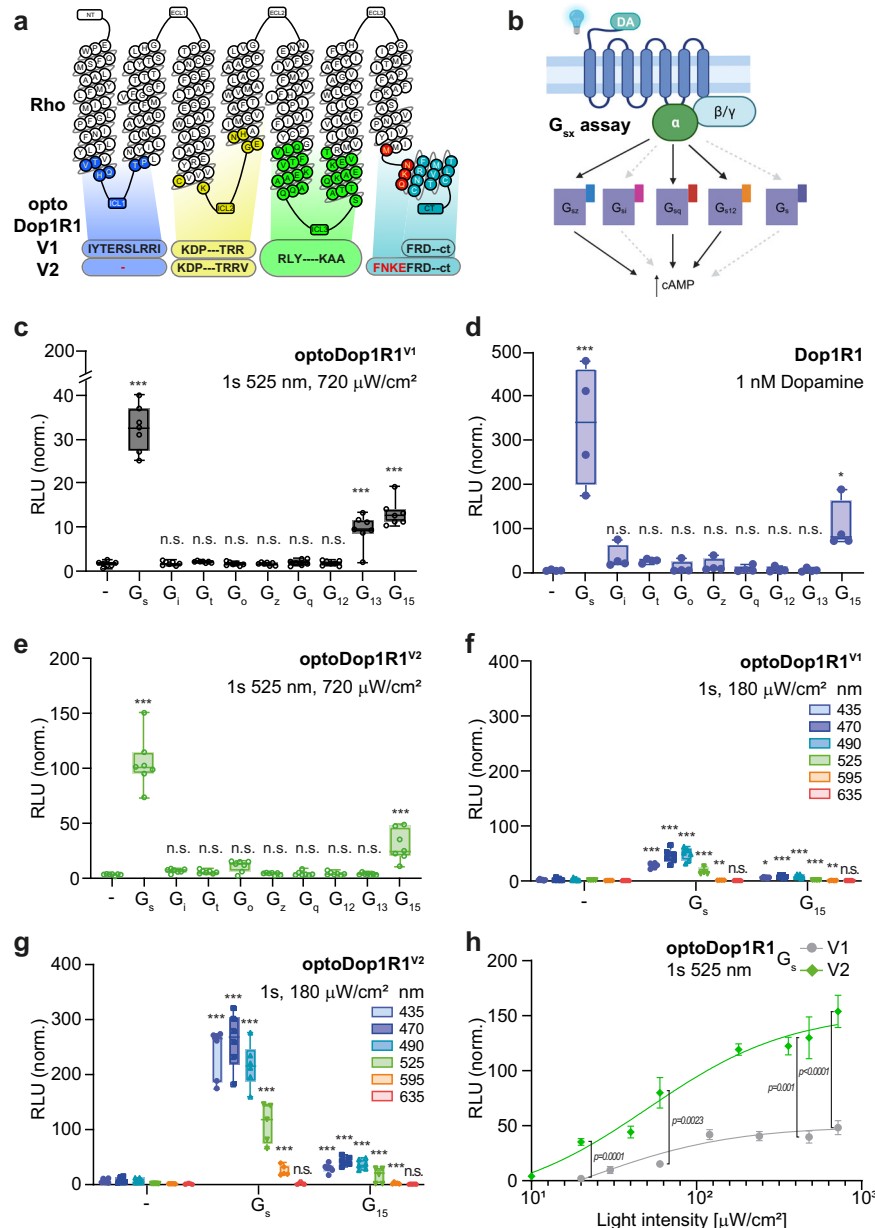

**Fig. 1 | Design and characterization of optoDop1R1$^{V2}$. a** Schematic overview of optoDop1R1 variants based on the original approach[14] (V1) and the optimized design (V2). **b** Schematic overview of the $G_{sX}$ assay. Coupling to chimeric $G_\alpha$ subunits ($G_{sX}$) redirects all G protein signaling to the same cellular response (cAMP). Created with BioRender.com. **c** G protein-coupling properties of optoDop1R1$^{V1}$ after activation with light (1 s, 525 nm, 720 μW/cm²). Maximum normalized responses are shown as relative light units (RLU, $n = 7$, **$p < 0.01$, ***$p < 0.001$, one-way ANOVA with Dunnett's post hoc test). **d** G protein-coupling properties of *Drosophila* Dop1R1 with 1nM dopamine. Maximum normalized responses are shown as relative light units (RLU, $n = 4$, *$p < 0.05$, ***$p < 0.001$, one-way ANOVA with Dunnett's post hoc test). **e** G protein-coupling properties of improved optoDop1R1$^{V2}$ after activation with light (1 s, 525 nm, 720 μW/cm²). Maximum normalized responses are shown as relative light units (RLU, $n = 7$, ***$p < 0.001$, one-way ANOVA with Dunnett's post hoc test). **f** Wavelength-dependent maximum

G protein activation of optoDop1R1$^{V1}$ after activation with light (1 s, 180 μW/cm², $n = 7$, *$p < 0.05$ **$p < 0.01$, ***$p < 0.001$, one-way ANOVA with Dunnett's post hoc test). **g** Wavelength-dependent maximum G protein coupling of optoDop1R1$^{V2}$ after activation with light (1 s, 180 μW/cm², $n = 6$, ***$p < 0.001$, one-way ANOVA with Dunnett's post hoc test). **h** Light intensity-dependent maximum of cAMP induction ($G_s$ coupling) of optoDop1R1$^{V1}$ and optoDop1R1$^{V2}$ after activation with light shown as relative light units (RLU, 1 s, 525 nm, mean ± SEM, optoDop1R1$^{V1}$: 20 μW/cm²: $n = 6$, 30/240 μW/cm²: $n = 3$; 60/480/720 μW/cm²: $n = 4$; 120 μW/cm²: $n = 8$; optoDop1R1$^{V2}$: 10/20/40/360 μW/cm²: $n = 6$, 60/720 μW/cm²: $n = 8$, 180 μW/cm²: $n = 4$; 480 μW/cm²: $n = 10$; $p$-values as indicated, unpaired two-tailed Student's $t$-test with Welch's correction. All $n$ indicate the number of independent experiments. All boxplots depict 75th (top), median (central line) and 25th (bottom) percentile, whiskers depict 99th (top) and 1st (bottom) percentile. Source data and statistical details are provided as a Source Data file.

redesigned the optoDop1R1 chimera and studied the effects of these changes.

## Characterization of Dop1R1 and optoDop1R1 activation profiles

We compared the activity of the *Drosophila* Dop1R1 receptor with its opto-variants designed with the previous (V1) or optimized (V2)

approach. Upon addition of dopamine, Dop1R1 showed strong coupling to $G_s$ as previously described[39], as well as $G_{15}$ and weak, not significant coupling to inhibitory G proteins (Fig. 1c, Supplementary Fig. 2a, b). $G_s$ and $G_{15}$ coupling showed dose-dependent responses in the nanomolar range (Supplementary Fig. 2b). In comparison, optoDop1R1$^{V1}$ activation using a 1 s light pulse (525 nm) resulted in $G_s$,

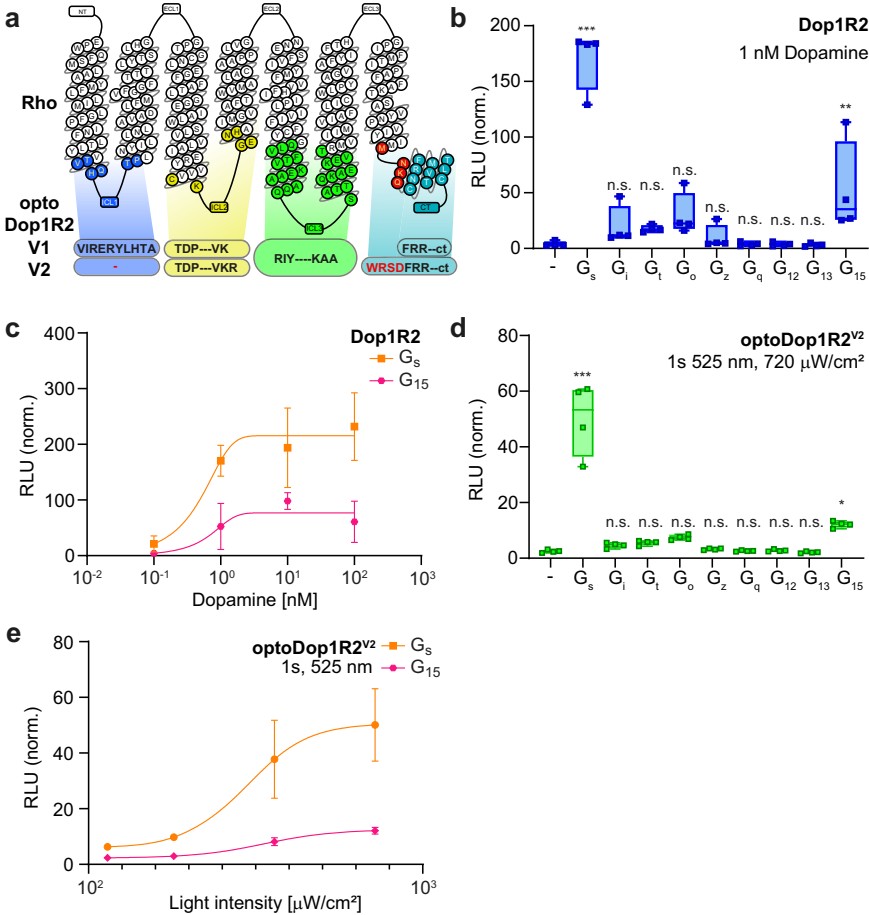

**Fig. 2 | Design and characterization of optoDop1R2$^{V2}$. a** Schematic overview of optoDop1R2$^{V2}$ design compared to V1. **b** G protein-coupling properties of *Drosophila* Dop1R2 with 1nM dopamine. Maximum normalized responses are shown as relative light units (RLU, $n = 4$, **$p < 0.01$, ***$p < 0.001$, one-way ANOVA with Dunnett's post hoc test). **c** DA concentration dependent maximum activation of $G_s$ and $G_{15}$ signaling of Dop1R2 (mean ± SEM, 0.1/10 nM: $n = 3$; 1.0/100 nM: $n = 4$). **d** G protein-coupling properties of optoDop1R2$^{V2}$ after activation with light (1 s, 525 nm, 720 µW/cm²). Maximum normalized responses are shown as relative light units (RLU, $n = 4$, *$p < 0.05$, ***$p < 0.001$, one-way ANOVA with Dunnett's post hoc test). **e** Light intensity-dependent maximum of $G_s$ and $G_{15}$ signaling induced by optoDop1R2$^{V2}$ (1 s, 525 nm, mean ± SEM, $n = 4$). All $n$ indicate the number of independent experiments. All boxplots depict 75th (top), median (central line) and 25th (bottom) percentile, whiskers depict 99th (top) and 1st (bottom) percentile. Source data and statistical details are provided as a Source Data file.

$G_{13}$ and $G_{15}$ coupling with moderate efficiency (Fig. 1c, Supplementary Fig. 2c). While significant induction of $G_s$ signaling was observed, the coupling profile did not match the Dop1R1 receptor profile entirely due to aberrant $G_{13}$ signaling and responses were comparatively small. In contrast, optoDop1R1$^{V2}$ activation more closely resembled the wild-type receptor displaying strong coupling to $G_s$ and $G_{15}$, as well as weak, not significant coupling to inhibitory G proteins (Fig. 1d, Supplementary Fig. 2d). As a previous report showed coupling of Dop1R1 to $G_q$[39], which was not observed in our experiments, we utilized the recently developed TRUPATH assay[48] allowing to directly measure G protein complex dissociation after receptor activation (Supplementary Fig. 2e). Using this independent approach, we confirmed the results of the $G_{sx}$ assay and observed $G_s$ and $G_{15}$ but not $G_q$ coupling of Dop1R1 and optoDop1R1$^{V2}$ under our conditions (Supplementary Fig. 2f, g). Of note, however, $G_{15}$ is a promiscuous $G_\alpha$ protein of the $G_q$ family able to induce $G_q$-type signaling via phospholipase C activation[49].

We then compared the wavelength-dependent $G_s$ and $G_{15}$ activation profiles of the two optoDopR variants. While maximum activation was observed with 470-490 nm light in cells expressing either receptor, optoDop1R1$^{V2}$ induced 5-10-fold higher responses than the corresponding V1 receptor (Fig. 1f, g, Supplementary Fig. 2h). In optoDop1R1$^{V2}$ expressing cells, strong $G_s$ activation was also observed in the green to orange wavelength range up to 595 nm, while it was weak in the case of optoDop1R1$^{V1}$. Direct comparison of light

intensity-dependent $G_s$ signaling induced by the two variants showed half-maximal activation at around 50 µW/cm² (at 525 nm) for both optoXRs (Fig. 1h).

However, responses elicited in the V2-expressing cells excelled in light sensitivity displaying 3- to 20-fold higher $G_s$ responses, particularly at low light intensities below 40 µW/cm². Overall, unlike the classic chimeric sequence-based approach, our optimized optoXR$^{V2}$ design yielded an optoDop1R1 variant exhibiting superior light sensitivity and high signaling specificity comparable to the Dop1R1 wild-type receptor.

**Generation and characterization of functional optoDop1R2$^{V2}$**

While for Dop1R1 both designs yielded active optoXRs albeit with different quality, the original approach did not produce a functional optoDop1R2 as no reliable light-dependent responses could be detected in the $G_{sx}$ assay (Supplementary Fig. 1a). We thus again turned to our optimized design and generated optoDop1R2$^{V2}$, which concordantly contained the Rho ICL1 and the extended C-terminus (Fig. 2a).

We first characterized *Drosophila* Dop1R2 using the $G_{sx}$ assay. Dop1R2 showed dose-dependent coupling to $G_s$, $G_{15}$ and inhibitory G proteins upon the addition of dopamine in the range of 0.1–100 nM (Fig. 2b, c, Supplementary Fig. 3a, b). Strikingly, in our optimized optoDop1R2$^{V2}$ the implemented changes indeed resulted

in a functional optoXR (Fig. 2d, e, Supplementary Fig. 3c). Similar to the wild-type receptor, optoDop1R2$^{V2}$ coupled to the same G proteins, prominently with $G_s$ and $G_{15}$ showing light-dose-dependent responses in the range of 114–720 $\mu W/cm^2$ (Fig. 2d, e). Furthermore, a similar light-dependent profile was also obtained for $G_i$ and $G_o$ responses (Supplementary Fig. 3d). The G protein-coupling profile and dose-dependent activity of optoDop1R2$^{V2}$ closely resembled the wild-type receptor in this assay, yet the maximum activation levels remained consistently lower under these conditions. As for optoDop1R1$^{V2}$, the rhodopsin-based optoDop1R2$^{V2}$ showed maximum responses to 470-490 nm light (Supplementary Fig. 3e). We also compared Dop1R2 and optoDop1R2$^{V2}$ responses in the TRUPATH assay. For both receptors, we observed comparable activation of $G_{15}$ but only minor induction of $G_s$ for optoDop1R2 suggesting favored activation of $G_q$-type signaling (Supplementary Fig. 3f, g). Overall, these results show that the optoXR$^{V2}$ design approach allowed the generation of functional and specific optoDopRs not obtainable with the previous strategy.

## Characterization of optoDopR localization in vivo

Based on the promising activity of optoDopRs$^{V2}$ in cell culture assays, we generated transgenes to investigate their functionality in vivo. We used the $\phi$C31 integration method to ensure comparable transgene expression efficiency due to the defined chromosomal integration site[50]. We first tested the expression and localization of optoDopRs in the *Drosophila* mushroom body (MB), the central learning and memory center in insects[51–54]. The principal MB neurons, Kenyon cells (KCs), receive olfactory and other sensory input via dendritic input at the calyx region. This information can then be modulated via compartmentalized dopaminergic innervation along their axonal arbors that are interconnected with MB output neurons (MBONs, Fig. 3a) to relay the information to other connected brain areas[54–57]. The expression of both Dop1R1 and Dop1R2 in KCs is required for learning and memory[34,37,39]. First, we expressed the optoDopRs in larval KCs and specific MBONs involved in odor-fructose association (MBON$^{g1/g2}$)[56] and investigated their cellular localization using immunohistochemistry. In larval KCs, the optoDop1R1$^{V1}$ signal was detectable in the soma and only weakly in axons and the calyx (Fig. 3b, Supplementary Fig. 4a). In comparison, optoDop1R1$^{V2}$ showed more prominent expression and was clearly visible in larval KC axons as well as in the calyx region (Fig. 3b, Supplementary Fig. 4a). Similarly, optoDop1R2$^{V2}$ showed prominent axonal and dendritic localization in larval KCs (Fig. 3b, Supplementary Fig. 4a). Quantitative analysis of axon/soma ratios of optoDopR signals demonstrated that the V2 variants had a more prominent axonal localization, while optoDop1R1$^{V1}$ was mostly confined to KC cell bodies (Fig. 3c). We then compared the localization of optoDop1R1$^{V2}$ in KCs to the localization of endogenous Dop1R1 visualized via a C-terminal split-GFP tag (Dop1R1$^{GFP_{11}}$), enabling cell type-specific endogenous labeling by co-expression of the complementary GFP (GPF$_{1-10}$) fragment[58]. In both cases, prominent expression was visible in the axonal lobes, calyx, and cell bodies (Fig. 3d). Quantitative analysis of compartmental signal intensity ratios revealed a similar distribution of endogenous Dop1R1 and optoDop1R1$^{V2}$ (Fig. 3e). We further compared their localization at the single cell level in MBON$^{g1/g2}$. We first confirmed the expression of endogenous Dop1R1 in these MBONs using the endogenous GFP tagging method (Fig. 3f). Dop1R1 localized to axon terminals and dendritic compartments in MBON$^{g1/g2}$. Again, unlike optoDop1R1$^{V1}$, optoDop1R1$^{V2}$ displayed a similar localization, including labeling of axonal varicosities resembling presynaptic sites (Fig. 3f, Supplementary Fig. 4b).

We obtained similar results for optoDopR localization in the adult MB with better expression levels for the V2 variants compared to optoDop1R1$^{V1}$, indicating more efficient folding, transport and/or stability of the improved versions (Supplementary Fig. 4c, d). Using an activity-induced expression system[59], we next analyzed the expression

of endogenous Dop1R1 as well as optoDopRs in individual adult KCs. Endogenous GFP-labeled Dop1R1 localized to somatodendritic compartments and was present within the axonal compartments of the MB lobes (Fig. 3g). Interestingly, Dop1R1 localized to presynaptic varicosities in KC axons, suggesting it exerts part of its function in presynaptic KC compartments (Fig. 3g, arrowheads). optoDop1R1$^{V2}$ again displayed a comparable localization, including labeling of axonal varicosities (Fig. 3h, arrowheads). In contrast, optoDop1R1$^{V1}$ was only weakly localized to axons and dendrites, labeling only a few axonal varicosities (Supplementary Fig. 4e). optoDop1R2$^{V2}$ prominently labeled axons and dendrites, suggesting efficient transport and localization to its site of action (Supplementary Fig. 4f). Overall, these data show that the V2 design yielded optoDopRs that are well expressed and, in case of optoDop1R1$^{V2}$, closely resemble endogenous receptor localization with prominent localization along KC/MBON axons including presynaptic sites.

## Characterization of optoDopR functionality in vivo

We next wanted to assay if 2nd messenger responses can be elicited by our optoDopRs in vivo. Dop1R1 has been reported to be primarily linked to $G_s$-dependent cAMP production, while Dop1R2 can induce intracellular calcium release via activation of $G_q$-family signaling that includes $G_{15}$[37,39,49]. Elevated cAMP and calcium levels in *Drosophila* larval nociceptors can elicit a stereotyped escape response[60], which we chose as a first proxy for functional activation of our optoXRs (Fig. 4a, b). We expressed optoDopRs in larval nociceptors and illuminated freely crawling larvae with blue light for 3 min. Similar to channelrhodopsins, functional optoXR expression requires retinal feeding as *Drosophila* does not produce sufficient amounts of *cis*- or all-*trans*-retinal to support the function of exogenously expressed light-sensitive GPCRs or channelrhodopsins, respectively. We expressed the blue light-activated adenylate cyclase bPAC[61] and the cation channelrhodopsin CsChrimson[62] as positive controls for cAMP and calcium-induced escape responses, respectively. bPAC and our optoXRs induced spontaneous rolling during light illumination, which generally occurred sporadically and with some delay (Fig. 4a, b, Supplementary Movies 1–4). In contrast, activation of CsChrimson resulted in a high percentage of animals rolling immediately after light onset (Supplementary Movie 5). Consistent with the predicted coupling to intracellular calcium stores by optoDop1R2, we also observed fast rolling responses in some cases. Overall, these data indicate that all optoXRs are capable of inducing 2nd messenger signaling in vivo with similarity to cAMP and calcium-induced escape responses.

To measure specific 2nd messenger responses induced by optoDopRs in vivo, we used fluorescent reporters for cAMP and calcium levels (Fig. 4a). Dop1R1 and Dop1R2 were previously shown to primarily regulate cAMP or store-released calcium levels in KC neurons, respectively[37]. We expressed the cAMP reporter Gflamp1[63] together with optoDop1Rs or bPAC in larval KCs and imaged light-induced cAMP changes in the soma and medial lobe regions in dissected live larval brains. bPAC activation with blue light was able to elicit strong cAMP increase, particularly in the KC soma region due to its cytosolic localization, and to a lesser extent also in the medial lobe region (Fig. 4c, d, Supplementary Movie 6). Similarly, activation of optoDop1R1$^{V1}$ resulted in a significant cAMP increase in the soma but not in the medial lobe region (Supplementary Fig. 5a–c). In comparison, activation of optoDop1R1$^{V2}$ resulted in cAMP increase preferentially in the medial lobe and to a lower degree in the soma region, which was largely dependent on the presence of 9-*cis*-retinal during the rearing of the animals (Fig. 4e–g, Supplementary Fig. 5d, Supplementary Movie 7). Axonal cAMP levels in the medial lobe decayed to background levels within approx. 60 s after a 10 s blue light stimulus. Of note, bPAC has been described to exhibit dark activity[64], and baseline fluorescence levels of Gflamp1 were significantly higher than for optoDop1R1$^{V2}$, suggesting optoDop1R1$^{V2}$ exhibits no or low dark

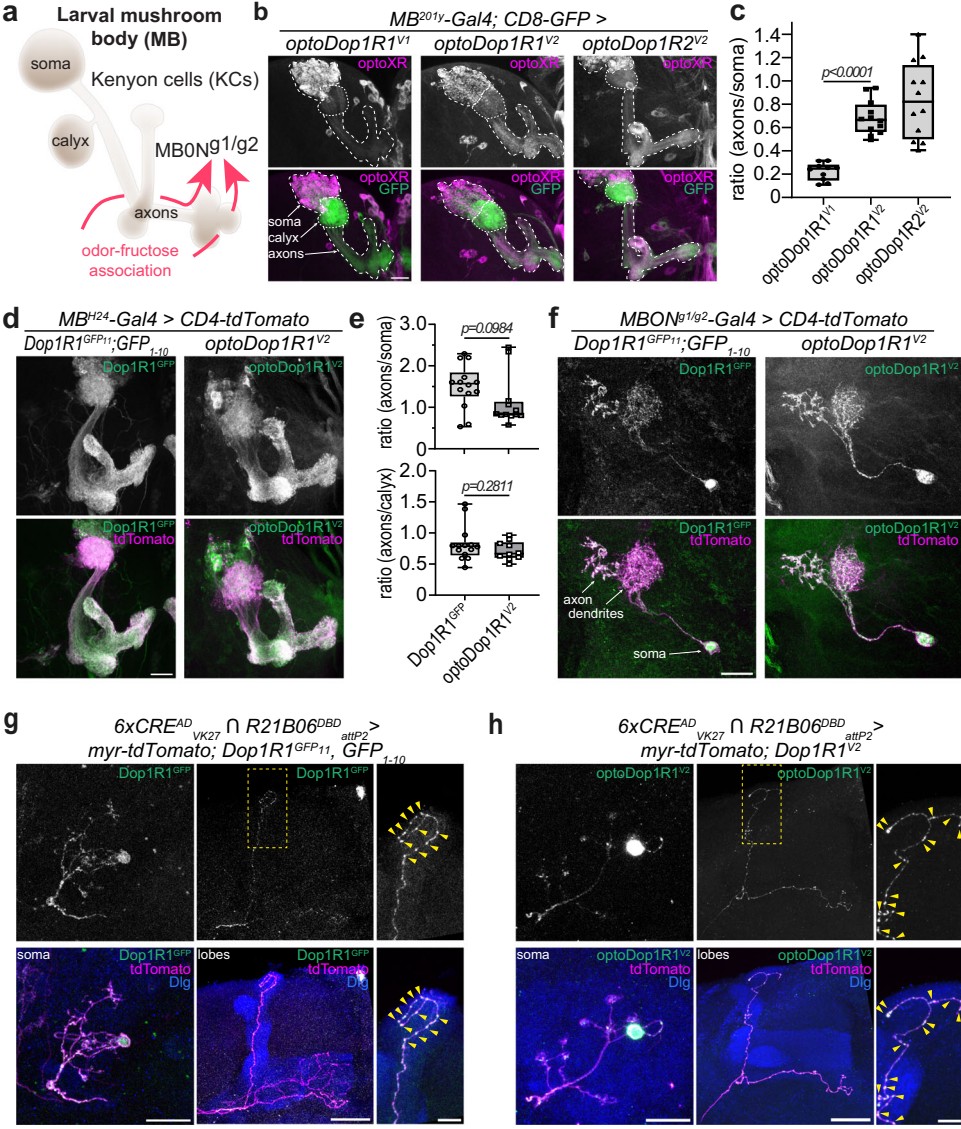

**Fig. 3 | In vivo localization of optoDopRs and endogenous Dop1R1. a** Schematic model of the larval mushroom body consisting of Kenyon cells (KCs) receiving input from dopaminergic neurons and connecting to output neurons (MBONs). Odor-Fructose association and learning require dopaminergic input and MBON$^{g1/g2}$ (adapted from ref. 56). **b** Immunohistochemistry of optoXRs (anti-Rho labeling) expression in KCs (labeled with CD8-GFP) in the larval mushroom body. Localization of KC somata, calyx (dendrites) and axons are outlined (scale bar: 25 µm). **c** Quantification of the optoDopR signal intensity ratios of axons/soma ($n = 10, 12, 12$ samples from 5, 6, 6 biologically independent animals, respectively, unpaired two-tailed Student's $t$-test). **d** Expression of endogenous Dop1R1 and optoDop1R1$^{V2}$ in the larval mushroom body (scale bars: 25 µm). **e** Quantification of the Dop1R1 and optoDop1R1 signal intensity ratios of axons/soma and axons/calyx. ($n = 14, 10$ samples from 7, 5 biologically independent animals, respectively, unpaired two-tailed Student's $t$-test). **f** Labeling of endogenous GFP-tagged Dop1R1 and expression of optoDop1R1$^{V2}$ (anti-Rho labeling) in MBON$^{g1/g2}$. Axon terminals, dendrites and soma are indicated and co-labeled by CD4-tdTomato expression

(representative image from two independent experiments with multiple samples). Scale bar: 20 µm. **g** Single-cell labeling of endogenous GFP-tagged Dop1R1 in adult KCs using activity-dependent induction of Gal4 activity[52]. Example of KC labeled with myristoylated(myr)-tdTomato and endogenous Dop1R1$^{GFP}$ (anti-GFP-labeled) in the somatodendritic region, axonal lobes and enlarged axon region (MB labeled by anti-Dlg). Presynaptic varicosities are indicated by arrowheads (representative images from two independent experiments with multiple samples). Scale bars: 10 µm, 20 µm, 5 µm. **h** Single-cell expression of optoDop1R1$^{V2}$ in adult MB showing a labeled KC expressing myr-tdTomato and optoDop1R1$^{V2}$ (anti-Rho labeled) displaying localization to the somatodendritic compartment, axonal lobes and enlarged axon region (MB labeled by anti-Dlg). Presynaptic varicosities are indicated by arrowheads (representative image from two independent experiments with multiple samples). Scale bars: 10 µm, 20 µm, 5 µm. All boxplots depict 75th (top), median (central line) and 25th (bottom) percentile, whiskers depict 99th (top) and 1st (bottom) percentile. Source data and statistical details are provided as a Source Data file.

activity compared to bPAC. In comparison, optoDop1R2$^{V2}$ activation resulted in weak and not significantly changed cAMP levels suggesting it has a limited capacity to regulate endogenous cAMP levels in KCs (Fig. 4f, g, Supplementary Fig. 5e, f).

We then tested for calcium store release upon optoDop1R$^{V2}$ activation by co-expression of the fluorescent calcium reporter GCaMP6s[65] in larval KCs. Activation of optoDop1R2$^{V2}$ resulted in robust calcium responses in the MB medial lobe and KC soma region

(Fig. 4h–j, Supplementary Fig. 5g, Supplementary Movie 8), consistent with the reported role of Dop1R2 in calcium store mobilization[37]. In contrast, optoDop1R1$^{V2}$ activation did not elicit significant calcium responses after blue light exposure suggesting it does not induce $G_q$-type signaling in KCs in vivo (Fig. 4i, j, Supplementary Fig. 5h, i). We also tested whether optoDopRs can be repeatedly activated under these conditions. optoDop1R1$^{V2}$ and optoDop1R2$^{V2}$ activation induced consistent cAMP and calcium responses during three

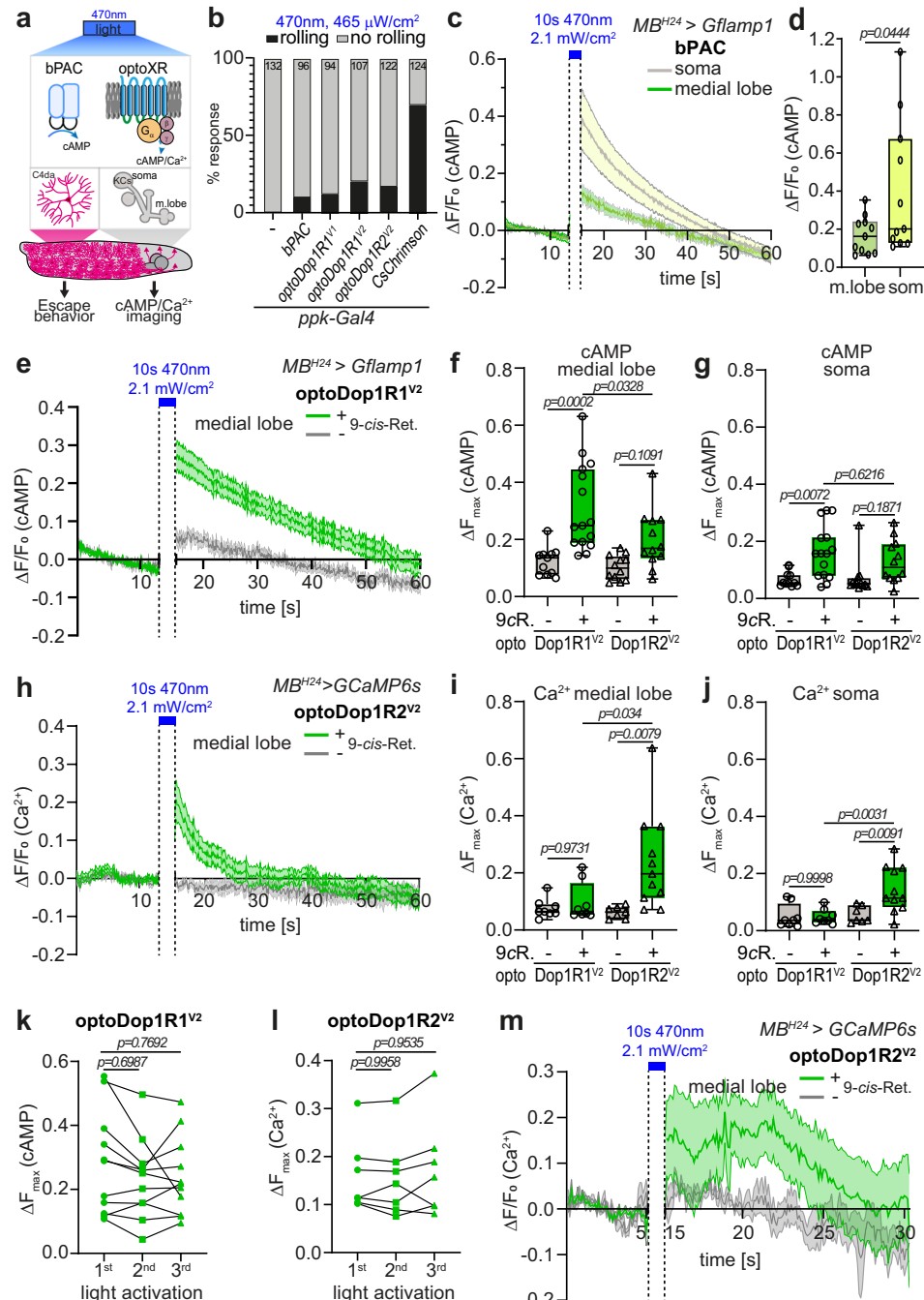

**Fig. 4 | In vivo characterization of optoDopR signaling activity. a** Schematic of of bPAC and optoXRs activation in larval nociceptors (C4da) or Kenyon cells (KCs). cAMP increase in C4da neurons elicits spontaneous larval escape responses. KC expression of GFlamp1 or GCaMP6s was used to image cAMP or Ca²⁺ responses, respectively. **b** Spontaneous escape responses (rolling) upon blue light illumination in larvae expressing bPAC, optoXRs, or CsChrimson in larval nociceptors (n animals as indicated). **c** cAMP responses over time in the larval mushroom body (soma and medial lobe) induced by bPAC activation (mean ± SEM, $n = 11$, 11 biologically independent samples). **d** Maximum cAMP responses in the KC soma and MB medial lobe after light-induced activation of bPAC ($n = 11$, 11 biologically independent samples, unpaired two-tailed Student's $t$-test). **e** cAMP responses in the medial lobe after optoDop1R1$^{V2}$ activation (mean ± SEM, $n = 11$, 15 biologically independent samples). **f** Maximum cAMP responses in the MB medial lobe after light-induced activation of optoDop1R1$^{V2}$ and optoDop1R2$^{V2}$ ($n = 11$, 15, 12, 12 biologically independent samples, one-way ANOVA with Tukey's post hoc test). **g** Maximum cAMP responses in the KC soma region after light-induced activation of optoDop1R1$^{V2}$ and optoDop1R2$^{V2}$ ($n = 11$, 15, 12, 12 biologically independent samples, one-way ANOVA with Tukey's post hoc test). **h** Calcium imaging in the larval mushroom body of

isolated brains using GCaMP6s and optoDop1R2$^{V2}$ with and without 9-cis-Retinal feeding (mean ± SEM, $n = 7$, 11 biologically independent samples). **i** Maximum calcium responses in the MB medial lobe after light-induced activation of optoDop1R1$^{V2}$ and optoDop1R2$^{V2}$ ($n = 8$, 8, 7, 11 biologically independent samples, one-way ANOVA with Tukey's post hoc test). **j** Maximum calcium responses in the KC soma region after light-induced activation of optoDop1R1$^{V2}$ and optoDop1R2$^{V2}$ ($n = 8$, 8, 7, 11 biologically independent samples, one-way ANOVA with Tukey's post hoc test). **k** Maximum cAMP responses in the MB medial lobe after repeated light-induced activation of optoDop1R1$^{V2}$ ($n = 9$ biologically independent samples, one-way ANOVA with Tukey's post hoc test). **l** Maximum calcium responses in the MB medial lobe after repeated light-induced activation of optoDop1R2$^{V2}$ ($n = 6$ biologically independent samples, one-way ANOVA with Tukey's post hoc test). **m** In vivo calcium imaging of the larval mushroom medial lobe using GCaMP6s and light-induced activation of optoDop1R2$^{V2}$ in animals reared with or without 9-cis-retinal (mean ± SEM, $n = 5$, 5 animals). All boxplots depict 75th (top), median (central line) and 25th (bottom) percentile, whiskers depict 99th (top) and 1st (bottom) percentile. Source data and statistical details are provided as a Source Data file.

consecutive activation cycles, respectively (Fig. 4k, l, Supplementary Fig. 5j, k).

We further confirmed optoDop1R2 activity by imaging light-induced changes in calcium levels in live intact larvae. Upon blue light illumination, we could detect calcium responses in the medial lobe as well as in KC somata (Fig. 4m, Supplementary Fig. 5l–n). Interestingly, calcium levels remained elevated for up to 10s after light stimulation, similar to the store release of calcium linked to dopaminergic activation in mammalian neurons[66]. Consistent with our imaging in dissected live larval brains, axonal responses in the medial lobe were overall stronger and more sustained than in the KC somata (Fig. 4i–m, Supplementary Fig. 5i–k), suggesting the local environment of receptor localization affects signaling efficiency.

Taken together, these data show that optoDopRs[V2] display the expected receptor type-specific signaling in KCs and that they can be repeatedly activated to induce relevant changes in cAMP and calcium levels in vivo.

## Functional analysis of dopaminergic signaling in fly larvae

We next wanted to test the functionality of the optoDopRs in relevant behaviors. Dopamine signaling plays a pivotal and conserved role in locomotion, reward, and innate preference behavior[2,3,5,31,67,68]. Dop1R1 function has been implicated in larval locomotion[42], and disruption of dopaminergic neuron function in flies and mammals results in locomotion defects and is a key feature of Parkinson's disease[69–72]. We used Rotenone-induced impairment of dopaminergic neurons in larvae, which resulted in reduced locomotion velocity and increased turning behavior as previously described[70] (Fig. 5a). We reasoned that locomotion deficits might be rescued by triggering dopaminergic signaling in the receiving cells. To this end, we expressed optoDopRs in the endogenous pattern of Dop1R1 using a knock-in Gal4 line (*Dop1R1[KO-Gal4]*). Locomotion of rotenone-treated larvae was tracked in the dark and subsequently upon green light illumination. We used green light (525 nm) in most of our assays due to strong innate avoidance responses toward blue light, which can interfere with behavioral readouts[73–75]. Expression and activation of optoDop1R1[V1] did not result in significant changes in locomotion and turning behavior in rotenone-treated larvae, except that green light induced an increase in turning behavior independent of optoDop1R1[V1] activity (Fig. 5b, Supplementary Fig. 6a). In contrast, we observed clear light-dependent recovery of locomotion using optoDop1R1[V2] activation (Fig. 5c, Supplementary Movie 9). Optogenetic activation of Dop1R1 signaling using the V2 variant significantly increased larval velocity and reduced the overall turning behavior of the Rotenone-treated animals, but not in control larvae without 9-*cis*-Retinal or Rotenone feeding (Supplementary Fig. 6b, c). This strongly suggests that optoDop1R1[V2] signaling in DA-receiving neurons can rescue toxin-induced dopaminergic impairment and corresponding locomotion deficits. Interestingly, expression and activation of optoDop1R2[V2] in the same pattern could also partially but not fully restore larval locomotion after Rotenone treatment (Fig. 5d, Supplementary Fig. 6d).

We next explored another core function of Dop1R1 signaling by addressing its function in learning and memory. *Drosophila* larvae are capable of reward learning, e.g., by forming olfactory preferences through odor-fructose association[38,53]. As in adult flies, the MB plays a key role in this process: KCs receive specific DAergic input and form a tripartite circuit with MB output neurons (MBONs), which together reinforce specific preference behavior[56]. As Dop1R1 signaling and cAMP increase in the MB are essential for learning in flies[33,34], we tested if optoDop1R1 activation during odor-fructose association can replace endogenous Dop1R1 function in KCs. We confirmed that KC-specific knockdown of Dop1R1 reduced learning performance in larvae (Supplementary Fig. 6e). Using optoDop1R1[V1] or optoDop1R1[V2] expression in KCs under these conditions partially rescued fructose-odor learning (Supplementary Fig. 6f, g). These results are consistent with the

reported function of Dop1R1 in learning and suggest that acute activation of optoDop1R1 signaling in KCs during odor-fructose association is sufficient for learning. Interestingly, even optoDop1R1[V1] activation could significantly rescue learning despite its weaker expression and predominantly somatic localization. However, as dopaminergic responses in KCs were shown to be compartmentalized within the axons[40,57], activation of optoDopRs in KCs cannot mimic this aspect of endogenous DA signaling. To avoid this issue, we tested for a potential function of Dop1R1 in MBON[g1/g2], which is specifically required for odor-fructose reward learning[56] and where we have shown endogenous Dop1R1 expression (see Fig. 3f). RNAi-mediated knockdown of Dop1R1 in MBON[g1/g2] indeed reduced larval reward learning strongly suggesting DA signaling via Dop1R1 has an essential modulatory function in these MBONs (Supplementary Fig. 6h, i). We additionally expressed optoDop1R1[V2] and activated it specifically during fructose-odor training, which partially rescued preference induction and learning compared to no light conditions (Fig. 5e, Supplementary Fig. 6j). This suggests that acute optoDop1R1[V2] activation during learning can functionally replace endogenous DA signaling in an MBON essential for odor-fructose association.

As DopR signaling is also involved in state and valence-dependent preference behavior[5], we further tested DopR knockout larvae in naïve odor preference. We focused on Amylacetate (AM) and 3-Octanol (3-OCT), two substances commonly used for larval odor-reward learning[76,77]. Dop1R1 knockout (*Dop1R1[ko-Gal4]*) and Dop1R2 knockout (*Dop1R2[ko-Gal4]*) larvae displayed no altered preference toward AM, which we used in our odor-reward learning paradigm (Supplementary Fig. 6k). However, *Dop1R2[ko]* larvae showed a specific reduction in 3-OCT preference (Fig. 5f). We therefore tested if optoDop1R2[V2] activation could rescue innate preference behavior. Light exposure during the preference assay indeed was able to restore 3-OCT preference in *Dop1R2[ko-Gal4]* larvae expressing optoDop1R2[V2] in an endogenous-like pattern (Fig. 5g). This result confirmed the functionality of optoDop1R2[V2] by restoring the in vivo function of its corresponding wild-type receptor in naïve odor preference.

## Functional analysis of dopaminergic signaling in adult flies

We further investigated the functionality of optoDopRs in adult flies, which requires very high light sensitivity of the optogenetic tools due to the low light penetrance of the fly cuticle, particularly below a wavelength of 530 nm[78]. We first tested the optoDop1R1[V2] function in the MB in an associative odor-shock learning paradigm, which requires dopaminergic input from PPL1 neurons to KCs[33,79]. We confirmed that Dop1R1 is required in KCs for odor-shock learning using an MB-specific RNAi-mediated knockdown (Fig. 6a, b). We then asked if activation of optoDop1R1[V2] in KCs can enhance performance when paired with the shock paradigm. We observed a trend toward more robust learning when optoDop1R1[V2] was activated during shock pairing, but this performance was not significantly enhanced (Fig. 6a, c). Interestingly, optoDop1R1 co-activation reduced trial-dependent variability in this assay, indicating more robust learning. We then asked if activation of DA signaling in KCs via optoDop1R1[V2] activation could replace the shock stimulus, which would imply that this artificial DA signaling could replace a teaching signal with a negative valence. However, optogenetic activation of DA signaling without the unconditioned stimulus did not confer any preference behavior (Fig. 6d). These results indicate that either activation of Dop1R1 signaling alone is not sufficient for associative preference behavior or that the missing restriction to a distinct KC compartment interferes with memory formation.

We then assayed DopR function in pigment dispersing factor (PDF) neurons, which consist of small (s-LN$_v$s) and large lateral ventral neurons (l-LN$_v$s). In particular, l-LN$_v$s are important for arousal, sleep and light input to the circadian clock[80,81], and previous studies suggested that Dop1R1 but not Dop1R2 has a depolarizing function in

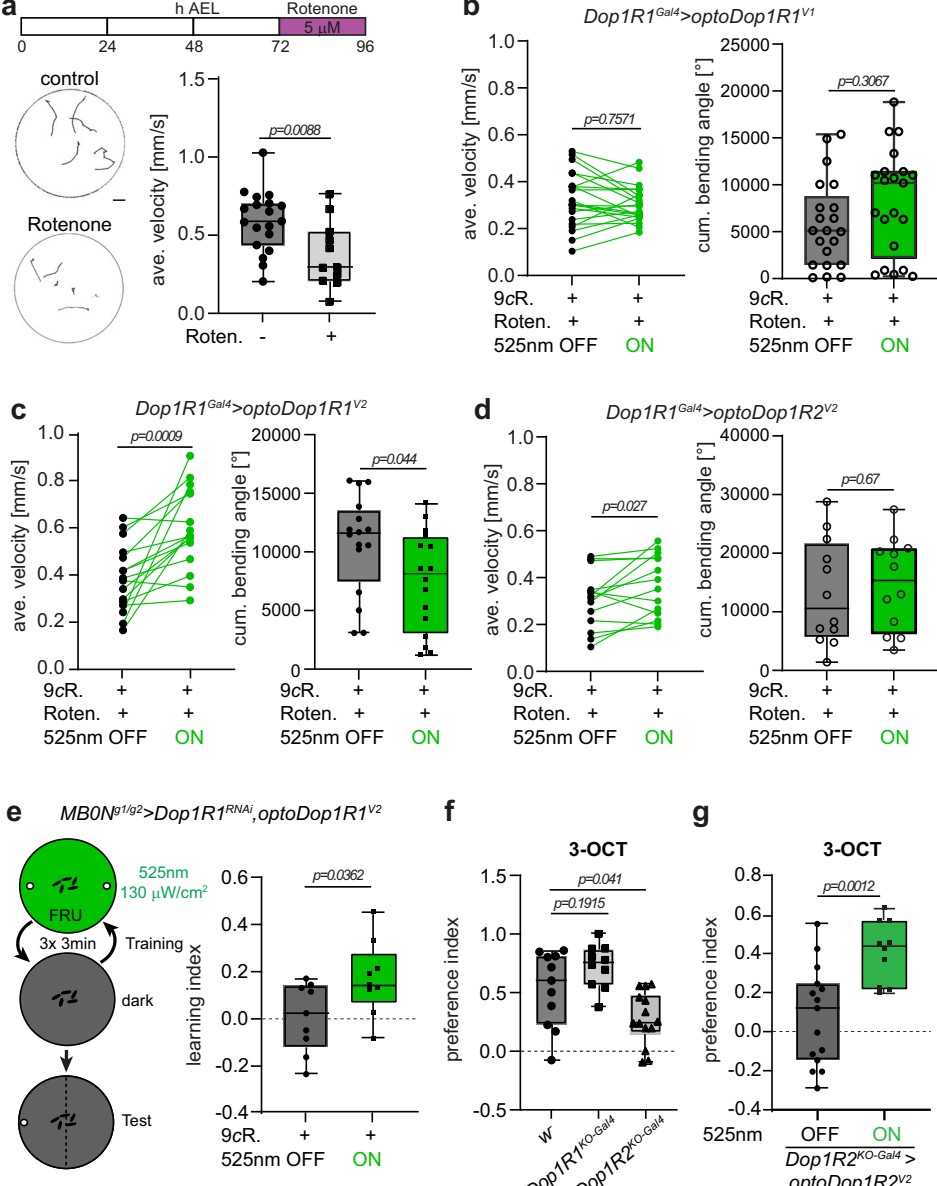

**Fig. 5 | Functional validation of optoDopRs in *Drosophila* larvae in vivo. a** Larvae were fed with 5 μM Rotenone for 24 h at 72 h after egg laying (AEL), inducing locomotion defects due to impaired dopaminergic neuron function. Representative larval tracks of control or Rotenone-fed animals are shown (1 min, scale bar: 10 mm). Quantification of the average velocity of control or Rotenone-fed animals ($n = 19$, 11 animals, two-tailed unpaired Student's *t*-test). **b** Average velocity and cumulative bending angles of larvae fed with 9-*cis*-Retinal (9cR) and Rotenone expressing optoDop1R1V1 in an endogenous Dop1R1 pattern (*Dop1R1ko-Gal4>optoDop1R1V2*) before and during 525 nm light illumination (1 min each, $n = 29$, 29 animals, two-tailed paired Student's *t*-test). **c** Average velocity and cumulative bending angles of larvae fed with 9-*cis*-Retinal (9cR) and Rotenone expressing optoDop1R1V2 (*Dop1R1ko-Gal4>optoDop1R1V2*) before and during 525 nm light illumination (1 min each, $n = 12$, 16 animals, two-tailed paired Student's *t*-test). **d** Average velocity and cumulative bending angles of larvae fed with 9-*cis*-Retinal (9cR) and Rotenone expressing optoDop1R2V2 (*Dop1R1ko-Gal4>optoDop1R2V2*) before and during

525 nm light illumination (1 min each, $n = 14$, 12 animals, two-tailed paired Student's *t*-test). **e** MBONg1/g2 and Dop1R1-dependent single odor-fructose learning in larvae. Animals expressing optoDop1R1V2 and Dop1R1RNAi in MBONg1/g2 were trained using fructose-odor learning (3x3min) with or without light activation during fructose exposure (3 min 525 nm, 130 μW/cm²). Learning index of 9cR-fed animals with and without light activation during training are shown ($n = 9$, 9 independent experiments, two-tailed unpaired Student's *t*-test). **f** Innate preference for 3-Octanol (3-OCT) in control (*w⁻*), *Dop1R1KO-Gal4* and *Dop1R2KO-Gal4* 3rd instar larvae ($n = 11$, 10, 14 independent experiments, one-way ANOVA with Tukey's post hoc test). **g** Innate preference for 3-OCT in *Dop1R2KO-Gal4* 3rd instar larvae expressing optoDop1R1V2. Innate preference for 3-OCT in 9cR-fed 3rd instar animals with and without light activation during the assay ($n = 15$, 10 independent experiments, two-tailed unpaired Student's *t*-test). All boxplots depict 75th (top), median (central line) and 25th (bottom) percentile, whiskers depict 99th (top) and 1st (bottom) percentile. Source data and statistical details are provided as a Source Data file.

l-LN_vs affecting the arousal state[82]. We assayed the activity of flies using the *Drosophila* Activity Monitor (DAM) system[83] from TriKinetics (Fig. 6e). Young flies were transferred to constant darkness after they had been reared under a 12 h dark/12 h light cycle. On the third day, darkness was interrupted by 12 arousing blue light pulses of different durations (10 min, 15 min, 20 min) given every hour for a period of 12 h

that was in phase with the previous light period. The blue light pulses not only efficiently aroused the flies but additionally activated opto-DopRs expressed in PDF neurons. Interestingly, expression and activation of optoDop1R1V2 were able to boost activity during the blue light periods compared to isogenic controls not fed with 9-*cis*-Retinal (Fig. 6f, g, Supplementary Fig. 7a, b). We performed a more detailed

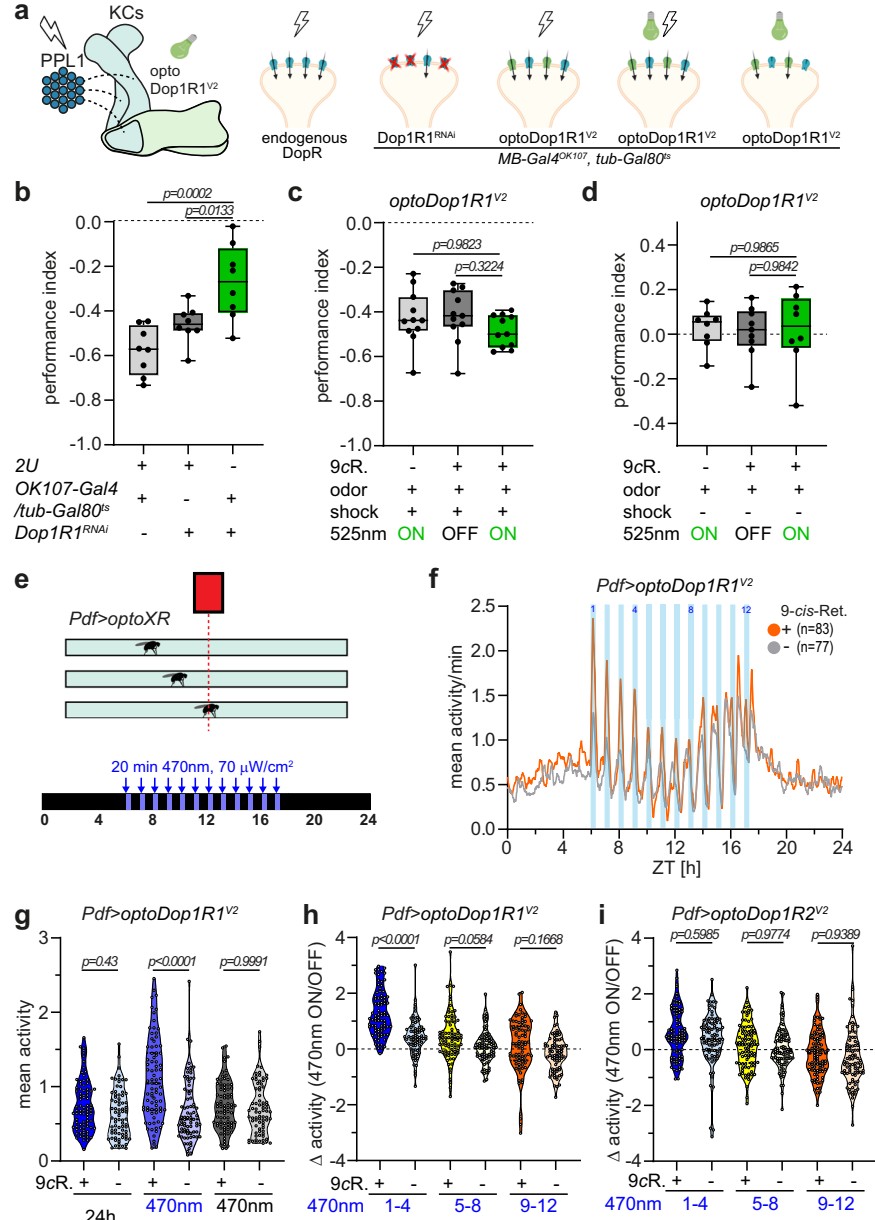

**Fig. 6 | Cell type-specific function of acute Dop1R1 activity in adult flies.**
**a** Schematic of adult mushroom body organization and associative odor-shock learning under different conditions (shock and/or light and altered Dop1R1 activity). Adult flies (*OK107-Gal4; tub-Gal80ts; UAS-RNAi/optoXR*) for experiments (**b**–**d**) were shifted to the permissive temperature (31 °C) 4 days prior to the behavioral assay to induce Gal4 expression. **b** Performance index after aversive odor-shock learning with or without adult-specific RNAi-mediated knockdown of Dop1R1 in Kenyon cells (*n* = 8 independent experiments, one-way ANOVA with Dunnett's post hoc test). **c** Performance index after aversive odor-shock learning with or without additional activation of optoDop1R1$^{V2}$ in Kenyon cells (*n* = 11 independent experiments, one-way ANOVA with Dunnett's post hoc test). **d** Performance index after pairing odor and optoDop1R1$^{V2}$ activation in Kenyon cells without shock (*n* = 8 independent experiments, one-way ANOVA with Dunnett's post hoc test).
**e** Schematic of activity monitor with flies expressing optoXRs in PDF neurons with daytime-dependent light activation using a blue light stimulus. The gray bar

indicates the flies' subjective day. **f** Mean activity during 24-h monitoring in flies expressing optoDop1R1$^{V2}$ in PDF neurons (mean, *n* = 83, 77 animals). Blue light pulses (12x 20 min, 1/h) during subjective daytime increase fly activity during the morning hours. **g** Mean activity of *Pdf>optoDop1R1$^{V2}$*-expressing flies during the entire 24 h, all light on and light off phases (*n* = 83, 77 animals, one-way ANOVA with Tukey's post hoc test). **h** Activity difference of flies expressing optoDop1R1$^{V2}$ in PDF neurons during light on/off times in the morning (1–4), midday (5–8) and afternoon (9–12) (*n* = 83, 77 animals, one-way ANOVA with Tukey's post hoc test). **i** Activity difference of flies expressing optoDop1R2$^{V2}$ in PDF neurons during light on/off times in the morning (1–4), midday (5–8) and afternoon (9–12) (*n* = 90 animals, one-way ANOVA with Tukey's post hoc test). All boxplots depict 75th (top), median (central line) and 25th (bottom) percentile, whiskers depict 99th (top) and 1st (bottom) percentile. All violin plots with single data points depict data distribution. Source data and statistical details are provided as a Source Data file.

analysis as the activity peaks were increasingly desynchronized with the blue light pulses (occurring after the light pulses) during the second part of the day. This revealed a significant effect of optoDop1R1$^{V2}$ activation specifically during the first 4h window (Fig. 6h). Next, we also tested optoDop1R2$^{V2}$ activation under the same conditions but did

not observe a significant effect on blue light-induced activity (Fig. 6i, Supplementary Fig. 7c–e). We then evaluated the expression of DopRs in l-LN$_v$s using respective Gal4 knock-in lines. We detected strong and specific reporter signal for Dop1R1 only in l-LN$_v$s, consistent with its function in light-induced arousal[82] (Supplementary Fig. 7f). In contrast,

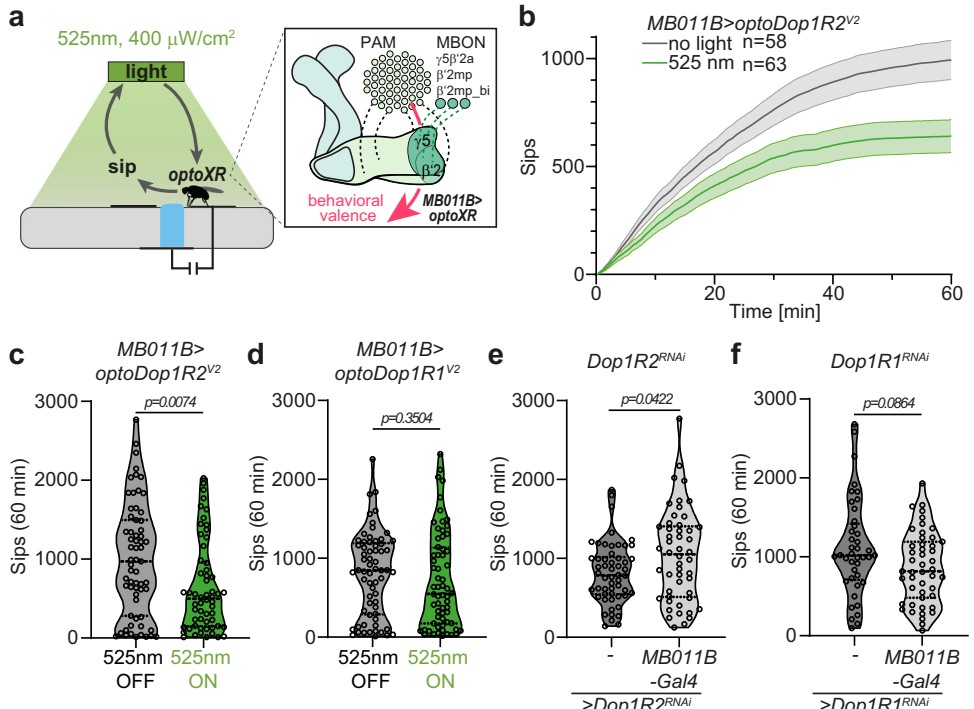

**Fig. 7 | Cell type-specific function of operant Dop1R2 activity in adult satiety.**
**a** OptoPAD setup allowing light stimulation upon feeding action. Flies expressing optoXRs in a subset of MBONs (MBONγ5β′2a,β2mp and β2mp-bilateral) related to behavioral valence receive a light stimulus (1 s 525 nm 400 µW/cm²) every time they feed on the sucrose drop. **b** Cumulative sips over time for flies expressing optoDop1R2$^{V2}$ using *MB011B-Gal4* without or with light stimulation (mean ± SEM, *n* = 58, 63 animals). **c** Total sips at 60 min for flies expressing optoDop1R2$^{V2}$ using *MB011B-Gal4* without or with light stimulation (*n* = 58, 63 animals, two-tailed Mann-Whitney test). **d** Total sips at 60 min for flies expressing optoDop1R1$^{V2}$ using

*MB011B-Gal4* without or with light stimulation (*n* = 65, 65 animals, two-tailed Mann-Whitney test). **e** Total sips at 60 min for flies expressing Dop1R2$^{RNAi}$ control or with *MB011B-Gal4* (*n* = 54, 50 animals, two-tailed Mann-Whitney test). **f** Total sips at 60 min for flies expressing Dop1R1$^{RNAi}$ control or with *MB011B-Gal4* (*n* = 41, 47 animals, two-tailed Mann-Whitney test). All boxplots depict 75th (top), median (central line) and 25th (bottom) percentile, whiskers depict 99th (top) and 1st (bottom) percentile. All violin plots with single data points depict data distribution, dotted lines depict 75th (top) and 25th (bottom) percentile, solid central line the median. Source data and statistical details are provided as a Source Data file.

the Dop1R2 reporter signal was very faint in l-LN$_v$s suggesting limited or no endogenous expression (Supplementary Fig. 7f). Together, these findings suggest a specific role for Dop1R1 signaling in l-LN$_v$s promoting morning activity upon arousal.

Finally, we also addressed a potential function of DopRs in adult MBONs previously implicated in encoding behavioral valence in MB-dependent tasks[52,84]. We chose an optoPAD setup which allows operant optogenetic stimulation of flies during feeding using a closed-loop system[85]. We expressed optoDopRs in relevant MBONs providing output of the γ5/β′2-compartments of the MB and activated DA signaling with green light pulses every time the flies were sipping food (Fig. 7a). Operant activation of optoDop1R2$^{V2}$ resulted in a decreased sipping rate over time suggesting that Dop1R2 signaling reduced the feeding drive and/or preference for the offered food (Fig. 7b, c). In contrast, operant optoDop1R1$^{V2}$ activation during feeding did not result in changed feeding behavior (Fig. 7d, Supplementary Fig. 8a). We further asked if the endogenous DopRs played a role in feeding in valence-encoding MBONs. RNAi-mediated knockdown of Dop1R2 but not Dop1R1 in MBON-γ5/β′2 resulted in an increased feeding rate (Fig. 7e, f, Supplementary Fig. 8b, c), suggesting a specific function for Dop1R2 in these MBONs in feeding-related behavior. Controls without expression of optoDopRs did not show altered feeding with or without operant light exposure (Supplementary Fig. 8d–g).

Taken together, operant optogenetic activation and RNAi-mediated decrease of Dop1R2 signaling in valence-encoding MBONs resulted in specific opposite effects on feeding. In contrast, manipulation of Dop1R1 activity in these MBONs did not alter feeding behavior. These findings strongly suggest that DA signaling in

valence-encoding MBONs regulates feeding drive specifically via Dop1R2. Overall, these data show neuron-specific functions of Dop1R1 and Dop1R2 signaling, which can be specifically induced by optoDopR activation.

## Discussion

By optimizing the chimeric optoXR approach, we generated highly functional and specific optoDopRs that allowed in vivo analysis of receptor-specific function and behavior in *Drosophila*. optoDop1R1$^{V2}$ showed enhanced and efficient activation in the blue and green spectral range (up to 595 nm) in cellular assays with light-dose-dependent activation properties resembling the wild-type receptor. While Rho-based optoXRs display a broad wavelength range of activation, they are compatible with red-shifted optogenetic tools, including channelrhodopsins like Chrimson that can be activated above 600 nm[62]. This should enable simultaneous optical control of neuronal activity via ion channel-mediated as well as neuromodulatory pathways, providing a way forward toward all-optical access to neuronal network function in vivo. For example, it will be highly interesting to combine optogenetic activation of specific DAergic neurons using CsChrimson as a teaching signal together with activation of Dop1R1 or Dop1R2 in KCs or responding MBONs to investigate timing-dependent synaptic plasticity and learning induced by receptor-specific signaling[37].

The high light sensitivity of the Rho backbone enables the activation of our optoXRs with blue or green light in adult flies in vivo despite less than 6% light penetrance of the adult cuticle in this spectral range[78,86]. Although Rho is known to inactivate after its light cycle and only slowly being recycled[87], we did not observe a run-down in

functionality in vitro or in vivo, possibly due to the abundance of the expressed optoXRs and the supplemented 9-*cis*-retinal.

Localization, cell type-specific and subcellular signaling dynamics are key to understanding endogenous GPCR signaling[24,88,89]. Recent evidence showed that 2nd messenger signaling can occur in nanodomains with receptor-specific profiles[90], emphasizing the importance of proper subcellular localization. Our optoDopR[V2]s display localization in the fly mushroom body in somatic and axonal compartments similar to their endogenous counterparts[58]. In contrast, the previous design did not yield a functional optoDop1R2[V1] receptor, and an optoDop1R1[V1] mostly localizing to the somatic compartment with a signaling profile different from the wild-type receptor. While some functional complementation was obtained with optoDop1R1[V1] in larval learning assays, unlike the V2 variants, it was not able to restore locomotion in animals with impaired DAergic neurons. This suggests that careful chimeric design is necessary to mimic endogenous receptor localization, signaling and function. This notion is consistent with optoDopR[V2]s mirroring the specific localization and signaling properties of their corresponding wild-type receptors. Dop1R1 has been shown to be required for cAMP responses in KCs, while Dop1R2 is required for calcium store release during olfactory conditioning[37]. Therefore, these tools will be beneficial to further unravel their temporal activation requirements to induce functional associations during learning or goal-directed behavior.

DA signaling plays a complex role in innate and adaptive behaviors. We used a wide range of behavioral paradigms showing that our optoDopRs exhibit cell type, receptor, and behavioral paradigm-specific functions in vivo. We showed that both optoDopR[V2]s are functional and can at least partially replace endogenous DopRs in several assays, including odor preference, locomotion and learning. At the same time, we uncovered a cell type-specific requirement of DopR signaling: only optoDop1R1[V2] but not optoDop1R2[V2] activation promoted LN[v]-mediated arousal; vice versa, operant activation of optoDop1R2[V2] but not optoDop1R1[V2] in valence-encoding MBONs was able to control feeding. DopR function has been extensively studied in KCs but has so far not been investigated in MBONs. Our findings therefore strongly suggest that corresponding MB outputs are also under the control of DA signaling. Thus, our optoXRs provide an entry point to gain insight into temporal and cell type-specific DA signaling requirements of the insect learning center, enabling detailed studies of the temporospatial requirement of DA signaling for learning, valence encoding, goal-directed and innate behavior in one of the most developed and heavily used model systems.

Although our improved optoXR design allowed the generation of optoDopRs that are functional in vivo, the complexity of GPCR signaling and the high sequence diversity of class A receptors make a general rational design of such tools difficult. Our incorporated adjustments provide an improved starting point that could be useful to generate optoXRs from other target receptors. Recently used approaches using structure-based design allowed improving the functionality of optoβ2AR, significantly increasing its light-induced signaling properties[17]. However, experimental structures of optoDopRs are currently not available. Similarly, the implementation of spectrally tuned or bistable rhodopsin backbones, as for example, shown for mouse Opn4[20,91], lamprey parapinopsin or mosquito Opn3[92], yields further promise to extend the optoXR toolbox. Combinations of these complementary methods could further improve optoXR design and functionality to enable efficient chimera generation allowing in vivo studies of other receptors in the future.

## Methods

### OptoDopR design

OptoDopR sequences were designed using Rho as the acceptor receptor, with segments containing G protein binding sites exchanged for those of the target receptor. To determine cut sites at the segment edges, a multiple protein sequence alignment of Rho and the target receptors was generated using Muscle[93]. Macros written in IgorPro were then used to cut and combine the aligned protein sequences in an automated fashion. V1 cut sites were based on previously published receptor designs[14,18]. For V2, cut sites around ICL1 and the C-terminus were amended to reflect previously published G protein binding sites[47]: residues in ICL1 were shown to not contribute to G protein binding, thus exchanges in ICL1 were omitted to retain the intact Rho ICL1. Conversely, the C-terminal cut sites were moved further toward the TM domains as these residues were shown to contribute to G protein binding. C-terminal Rho residues (TETSQVAPA) were added to the C-terminus of optoXRs V1/V2 chimeric constructs to enable comparative immunolabeling using anti-Rho antibodies. Protein sequences of chimeric GPCRs generated in this study are shown in Supplementary Table 2.

### Plasmids

cDNAs of wild-type *Drosophila* Dop1R1 and Dop1R2 were obtained from the *Drosophila* Genomics Resource Center (DGRC, Bloomington, IN, USA) and cloned into pCDNA3.1 (Thermo Fisher). optoDop1R1 and optoDop1R2 chimera (V1 and V2) were synthesized as codon-optimized cDNAs (Thermo Fisher) and cloned into pCDNA3.1 and pUAttB. Chimeric G proteins for the G[sx] assay[45] and the TRUPATH assay plasmids were obtained from Addgene (Watertown, MA, USA).

### Cell culture and live-cell G protein-coupling assays

G protein coupling of wild-type and chimeric GPCR constructs was tested in HEK293T cells (gift from M. Karsak, ZMNH, University Medical Center Hamburg-Eppendorf, Germany) or HEK293-ΔG7[46] (lacking GNAS/GNAL/GNAQ/GNA11/GNA12/GNA13/GNAZ; gift from A. Inoue, Tohoku University, Japan) using the G[sx] assay[45]. The GPCR constructs were subcloned into pcDNA3.1 (Thermo Fisher). HEK293T cells were incubated in DMEM medium containing 10% FBS (PAN Tech.) with penicillin (100 U/mL) and streptomycin (100 mg/mL) at 37 °C and 5% $CO_2$. For transfection, cells were seeded into white 96-well plates (Greiner Bio One) coated with poly-L-lysine (Sigma Aldrich). On the next day, the medium was changed to DMEM/FBS containing 10 mM 9-*cis*-Retinal.

Cells were then transfected with individual opto- or wild-type receptors, G protein chimera (G[sx]) and Glo22F (Promega) using Lipofectamine 2000 (Thermo Fisher). Cells were incubated at 37 °C and 5% $CO_2$ for 24 h and the medium was replaced with Leibovitz's L-15 media (without phenol-red, 1% FBS) containing 2 mM beetle luciferin (in 10 mM HEPES pH 6.9) and 10 mM 9-*cis*-retinal (for optoXRs) and cells were incubated at room temperature for 1h. For optoXR experiments, the plates were kept in the dark at all times before illumination and cAMP-dependent luminescence was measured using a Berthold Mithras multimode plate reader (Berthold Tech., Germany). Baseline luminescence was measured three times, and activation of DopRs was induced by ligand addition (dopamine at various concentrations diluted in L-15). For optoDopR activation, cells were illuminated with a 1-s light pulse using an LED light plate (Phlox Corp., Provence, France) or a CoolLED pE-4000 (CoolLED, Andover, UK). Specific light intensities and wavelengths are indicated in individual experiments. Technical duplicates were performed for all experiments with a minimum of three independent trials. For data quantification, each well was normalized to its pre-activation baseline.

For the TRUPATH assay[48], HEK293ΔG7 cells were seeded as described above, co-transfected with RLuc8-G[α], G[β], G[γ]-GFP2 and wild-type or opto-DopRs in a 1:1:1:1 ratio (100 ng/well total DNA) using Lipofectamine 2000. Cells were incubated for 24 h at 37 °C, 5% $CO_2$ and subsequently, in Leibovitz's L-15 media (without phenol-red, with L-glutamine, 1% FBS, penicillin/streptomycin 100 mg/ml) and 9-*cis*

**Table 1 | Transgenic *Drosophila* lines used in this study**

| Line | Label | Source |
|------|-------|--------|
| *Dop1R1*[KO-Gal4] | Knockout-Gal4 of Dop1R1 | BDSC# 84714 |
| *UAS-Dop1R1*[RNAi] | Knockdown of Dop1R1 | BDSC# 62193 |
| *UAS-Dop1R2*[RNAi] | Knockdown of Dop1R2 | BDSC# 51423 |
| *Dop1R2*[KO-Gal4] | Knockout-Gal4 of Dop1R2 | BDSC# 84715 |
| *201y-Gal4* | Expresses GAL4 in the mushroom body | BDSC 64296 |
| *H24-Gal4* | Expresses GAL4 in the mushroom body | BDSC# 51632 |
| *UAS-bPAC* | Optogenetic cAMP induction | Stierl et al. (ref. [61]), BDSC# 78788 |
| *UAS-optoDop1R1*[V2] | Optogenetic Dop1R1 activation | This study |
| *UAS-optoDop1R2*[V2] | Optogenetic Dop1R2 activation | This study |
| *UAS-optoDop1R1*[V1] | Optogenetic Dop1R1 activation | This study |
| *ppk-Gal4* | Expresses GAL4 in C4da neurons | Han et al. (ref. [99]) |
| *UAS-CsChrimson-GFP* | Optogenetic activation | Klapoetke et al. (ref. [62]), BDSC# 55136 |
| *UAS-Gflamp1* | cAMP reporter | Wang et al. (ref. [63]) |
| *UAS-Gcamp6s* | calcium reporter | Chen et al. (ref. [65]) |
| *MBONg1g2-Gal4* | Expresses GAL4 in MBON-g1,g2 | Saumweber et al. (ref. [56]) |
| *Pdf-Gal4* | Expresses GAL4 in l-LN$_v$ and s-LN$_v$ | BDSC# 6899 |
| *MB011B-Gal4* | Expresses GAL4 in valence-encoding MBONs | Aso et al. (ref. [84]) |
| *2U* | w[1118] (isoCJ1) Canton-S derivative | Tully et al. (ref. [100]) |
| *OK107-Gal4* | Expresses GAL4 in the mushroom body | BDSC# 854 |
| *tub-Gal80*[ts] | Expresses temperature sensitive GAL80 in all cells | BDSC# 7019 |
| *R21B06-splitGal4*[DBD] | Expresses GAL4[DBD] in the mushroom body | Aso et al. (ref. [52]) |
| *6xCRE-splitGal4*[AD] | Expresses GAL4[AD] in a Cre-dependent manner, VK27 insertion | This study, see Siegenthaler et al. (ref. [59]) |
| *UAS-myr::tdTomato* | Fluorescent reporter line | Pfeiffer et al. (ref. [101]), BDSC# 32223 |
| *UAS-Dop1R1GFP₁₁, UAS-spGFP₁₋₁₀* | Dop1R1 knock-in line with C-terminal GFP$_{11}$ tag | Kondo et al. (ref. [58]) |
| *10xUAS-myr::GFP* | Fluorescent reporter line | Pfeiffer et al. (ref. [101]), BDSC# 32197 |

retinal (10 μM) and kept in the dark. For performing BRET assays, the medium was changed to HBBS, supplemented with 20 mM HEPES, 10 μM 9-*cis*-retinal and 5 μM Coelenterazine 400a, and incubated for 5 min at RT. optoDopRs were activated using a 1 s,470 nm light pulse (collimated CoolLED pE4000, Andover, UK). Native DopRs were activated by injection of DA with a final concentration of 1 μM. BRET ratio changes were determined from RLuc8-G$_\alpha$ and G$_\gamma$-GFP2 emission using a Berthold Mithras multimode plate reader with BRET2 filters (410m80/515m40, Berthold Tech.) over a 90s timeframe directly after light or DA application.

### *Drosophila melanogaster* stocks

All *Drosophila* stocks were raised and treated under standard conditions at 25 °C and 70% relative humidity with a 12 h light/dark cycle on standard fly food unless stated otherwise. Transgenic UAS-optoDopR lines were generated by phiC31-mediated site-specific transgene using the attP2 site on the 3rd chromosome (FlyORF Injection Service, Zurich, Switzerland). Stocks were obtained from the Bloomington (BDSC) *Drosophila* stock centers unless otherwise noted. We used the lines as shown in Table 1.

### Immunochemistry

Larval brains from 3rd instar animals (96 h ± 3 h AEL) of the indicated genotypes were dissected in phosphate-buffered saline (PBS) and fixed for 15 min at room temperature in 4% paraformaldehyde/PBS, washed in PBST (PBS with 0.3% Triton X-100) and incubated in 5% normal donkey serum in PBST. OptoDopR expression was analyzed using a mouse anti-Rho antibody detecting the C-terminal Rho epitope present in all optoXRs (1D4, Cat #MA1-722, 1:1000, Thermo Fisher, CA,

USA) at 4 °C overnight, washed in PBST 3 times (5 min in each time) and incubated with secondary antibodies (donkey anti-mouse Alexa 488 Cat #715-545-150, Jackson Immunoresearch, or goat anti-mouse Alexa 546 Cat # A-11030, Thermo Fisher, CA, USA, 1:300) for 1h. After washing, samples were mounted on poly-L-lysine coated coverslips in Slow Fade Gold (Thermo Fisher, CA, USA). Native reporter fluorescence was sufficiently bright to be visualized together with antibody immunostaining by confocal microscopy (Zeiss LSM900AS2, Zeiss, Oberkochen, Germany). Confocal Z-stacks were processed in Fiji (ImageJ, NIH, Bethesda, USA).

Adult brains of 3- to 7-day-old flies of the indicated genotypes were dissected in hemolymph-like saline (HL3) and fixed for 1 h at room temperature in 2% paraformaldehyde/HL3. After washing in PBST (PBS with 0.5% Triton X-100) and incubation in 5% normal goat serum in PBST, samples were incubated with mouse anti-Rhodopsin (1D4, Cat #MA1-722, 1:1000, Thermo Fisher) to detect optoDopR expression, rabbit anti-DsRed (1:2000, Cat #632496, Takara Bio Inc.), mouse anti-GFP (1:2000, Cat #A-11120, Thermo Fisher), rabbit or guinea pig anti-Discs large (Dlg, 1:30000 and 1:1000;[94]) antibodies for 4 h at room temperature, followed by 2 nights at 4 °C. For DopR/PDF co-expression analysis in adult brains, mouse anti-PDF (Cat #PDF C7, 1:1000, DSHB) and chicken anti-GFP (Cat #ab13970, Abcam, 1:2000) were incubated for 24h at 4 °C. Samples were subsequently washed in PBST (3 x 30 min) and incubated with secondary antibodies (goat anti-mouse Alexa 488 Cat # A-11001, goat anti-rabbit Alexa 594 Cat # A-11012, goat anti-guinea pig Alexa 647 Cat # A-21450, 1:1000, Thermo Fisher) for 4 h at room temperature, followed by 2 nights at 4 °C. For DopR/PDF co-expression analysis, secondary antibodies (donkey anti-mouse Alexa 555 Cat # A-31570, 1:400; goat anti-chicken Alexa 488 Cat

# A-11039, 1:200, Thermo Fisher) were incubated for 6h at room temperature. After washing, a pre-embedding fixation in 4% paraformaldehyde/PBS was performed for 4 h at room temperature. Samples were washed in PBST (4 × 15 min) followed by 10 min in PBS. Brains were mounted on poly-L-lysine coated coverslips. An ethanol dehydration series and a xylene clearing series were performed and the samples were mounted in DPX[95]. Images were taken on a Leica STELLARIS 8 confocal microscope using a 20x (NA 0.75) and 93x (NA 1.3) glycerol immersion objective. Confocal z-stacks were processed in Fiji (ImageJ, NIH, Bethesda, USA).

## Calcium and cAMP imaging in *D. melanogaster* larvae
3rd instar larval brains (96 h ± 3 h AEL) were partially dissected in physiological saline buffer (108 mM NaCl, 5mM KCl, 2mM CaCl$_2$, 8.2 mM MgCl$_2$, 4 mM NaHCO$_3$, 1 mM NaH$_2$PO$_4$, 5 mM trehalose, 10 mM sucrose, 5 mM HEPES, pH 7.5) and mounted on poly-L-lysine-coated cover slips in the saline buffer with or without 5mM 9-*cis*-Retinal (for opto-Dop1R1 and opto-Dop1R2). Gflamp-1 or GCaMP6s was utilized to monitor cAMP or calcium levels, respectively. Live imaging of Kenyon cell somata and medial lobes expressing Gflamp-1 or GCaMP6s in the mushroom body was performed using confocal microscopy with a 40x/NA1.3 objective (Zeiss LSM900AS2, Zeiss, Oberkochen, Germany). OptoDopR[V2] or bPAC activation was achieved using a 470 nm LED light with an intensity of 2.10 mW/cm². Confocal time series were recorded at 7.5 frames/s (128 × 128 pixels, 600 frames total or 1000 frames total for repeated light activation). KC somata or medial lobes were focused, and after a stable imaging period of 100 frames, the 470 nm LED was activated for 10 s. Confocal time series were analyzed using image registration (StackReg plugin, ImageJ) to correct for XY movement, and Gflamp-1 signal intensity in the soma and medium lobe was quantified using the Time Series Analyzer V3 plugin (ImageJ). Baseline (F$_0$) was determined as the average of 95 frames before activation. The relative maximum intensity change (ΔF$_{max}$) of Gflamp-1/GCaMP6s fluorescence was calculated after normalization to baseline.

Live imaging of calcium responses in intact 3rd instar larvae was performed under low light conditions. Larvae were mounted in 90% glycerol, sandwiched between a coverslip and the slide with the aid of silicon paste. Calcium responses were recorded from the soma/calyx region and the medial lobe of the mushroom body using *UAS-GCaMP6s* and *UAS-OptoDop1R2* [V2] under the control of *H24-Gal4*. Animals were reared in the dark on grape agar plates supplemented with yeast paste and 9-*cis*-retinal. The soma, as well as the medial lobe of the mushroom body, were live imaged using a Zeiss LSM 780 2-photon microscope and a 25x/NA1.0 water immersion objective. For activation of the optoDop1R2[V2], larvae were subjected to 10s blue light stimulation (470 nm, 720 μW/cm², CoolLED) twice with an interval of 30s between each pulse. Only datasets without significant Z-drift were used for analysis. Analysis of the time series was performed using Fiji (ImageJ, NIH, Bethesda, USA) as described above. Normalized calcium responses were obtained by subtracting the amplitude of the pre-stimulation baseline (average of 50 frames) from the stimulation evoked amplitude. The calcium response was recorded before and after the light stimulus due to PMT overexposure during the light pulse. Graphs showing the mean ± s.e.m were generated with GraphPad Prism (GraphPad, San Diego, CA, USA). Boxplots were used to show the comparison between the maximum responses (ΔF$_{max}$/F$_0$) and analyzed with unpaired Student's *t*-test with Welch's correction.

## cAMP-induced nociceptive behavior in *D. melanogaster* larvae
For cAMP-induced nociceptive behavior, larvae expressing UAS-bPAC, UAS-CsChrimson or UAS-optoDopRs under the control of *ppk-Gal4* were staged and fed in the dark on grape agar plates (2% agar) with yeast paste containing 5 mM 9-*cis*-retinal (optoXRs) or all-*trans*-retinal (CsChrimson). Staged 3rd instar larvae were placed on a 1% agar film on an FTIR (frustrated total internal reflection) based tracking system (FIM, University of Münster) with 1ml water added. Experiments were performed under minimum light conditions (no activation). After 10 s, larvae were illuminated with 470 nm light (465 μW/cm²) for 3 min. Behavioral responses during the 3 min were recorded and categorized as rolling (full 360° rotation along the larval body axis) or no rolling (incomplete rolling, bending, turning, or no response). Each animal was counted only once, and the cumulated categorized responses were plotted as a contingency graph. Staging and experiments were performed in a blinded and randomized manner.

## Locomotion assays in *D. melanogaster* larvae
*D. melanogaster* larvae were staged in darkness on grape agar plates containing yeast paste with or without 5 mM 9-*cis*-retinal. For the indicated experiments, larvae were additionally fed with Rotenone for 24 h at 72 h after egg laying (AEL) to impair dopaminergic neuron function. Third instar larvae (96 h ± 4 h AEL) were used for all experiments. Animals were carefully selected and transferred under minimum red-light conditions to a 1% agar film on an FTIR (frustrated total internal reflection) based tracking system (FIM, University of Münster). Five freely moving larvae per trial were video-captured and stimulated with 525 nm light (130 μW/cm²) for activation of optoDop1R1[V2]. Animal locomotion was tracked with 10 frames/s for up to 120s. For locomotion analysis, velocity and bending angles were analyzed using the FIMtrack software (https://github.com/kostasl/FIMTrack). Only animals displaying continuous locomotion before the light stimulus were analyzed. Average locomotion speed and cumulative bending angles were analyzed and plotted for the first 30 s under dark or light conditions.

## Innate odor preference and olfactory behavior assays in *D. melanogaster* larvae
Groups of 20 staged mid-3rd instar larvae (96 h ± 4 h AEL) were placed in the middle of a 2% agar plate containing a container with 10 μl n-amylacetate (AM, diluted 1:50 in mineral oil; SAFC) or 3-Octanol (3-Oct, Sigma) on one side and a blank on the other side. For rescue experiments, assays were performed either in the dark or using light conditions (525 nm, 130 μW/cm²) during the preference behavior. Assays were video-captured for 5 min under infrared light illumination to monitor larval distribution with a digital camera (Basler ace-2040 gm, Basler, Switzerland). After 5 min, the number of larvae on each side was determined and the odor preference was calculated as (n(larvae) on odor side − n(larvae) on blank side)/total n(larvae).

## Odor-fructose reward learning assays in *D. melanogaster* larvae
Odor-fructose reward learning was performed essentially as described[77]. Groups of 20 larvae each were placed in a petri dish coated either with plain 1% agar or 1% agar with 2 M fructose as a reward in the presence of 10 μl n-amylacetate (AM, 1:50). The odor-reward or no reward pairing was done for 3 min (or 5 min; as indicated in experiments), alternating 3x between training (odor⁺), while the unpaired group received odor and reward during separate 3 min (or 5 min as indicated) training (blank⁺). For all optogenetic lines, training was performed under minimum red-light conditions or with 525 nm light activation (130 μW/cm²) during fructose reward training. Reciprocal training was performed for all genotypes and conditions (blank/odor⁺ and blank⁺/odor, respectively).

After three training cycles, larval preference toward the trained odor (AM or blank) was recorded in darkness using a Basler ace-2040gm camera (same setting as for the olfactory behavior assay). The number of larvae on each side was calculated after 5 min, and odor preferences were calculated for the paired and unpaired groups. The learning index (LI) was then calculated using the following formula:

$$LI = (Odor - Pref_{Paired} \text{--} Odor - Pref_{Unpaired})/2 \qquad (1)$$

## Odor-shock learning behavior assays in *D. melanogaster* adult flies

Aversive olfactory conditioning of adult flies was performed as described before[77]. Conditioning was performed in the dark at 21 °C and 75% humidity using 3- to 7-day-old flies. Groups of flies were loaded into custom-made copper grid tubes with high-power LEDs mounted at the end of the tube (525 nm, Ø 37 µW/mm²). Flies were exposed to a constant air stream or the odorized air stream (750 ml/min).

Experimental flies were raised at 20 °C and shifted to 31 °C four days prior to the experiments to induce Gal80[ts]/Gal4-dependent gene expression. Flies were transferred to 0.4 mM 9-*cis*-retinal food ~48 h prior to the experiment and kept in the dark.

For conditioning the odors 4-MCH (1:250, Merck, Darmstadt, Germany, CAS #589-91-3) and 3-OCT (1:167, Merck, Darmstadt, Germany, CAS #589-98-0) were diluted in mineral oil (Thermo Fisher, Waltham, MA, CAS #8042-47-5). Flies were conditioned following a five times spaced training paradigm. After a resting period of 3 min with only airflow the flies were exposed to the stimuli as indicated in the figure. The CS[+], electric shocks (twelve 1.5-s 90 V shocks with 3.5-s intervals) (Fig. 5b) and pulsed green light (4 Hz, 0.125 s on and 0.125 s off) (Fig. 5c, d) were applied simultaneously for 60 s. After 45 s of airflow, the CS[–] was presented for 60 s. This training cycle was repeated five times with 15-min breaks in between cycles. Odors for CS[+] and CS[–] were interchanged for each *n*.

Learning behavior was subsequently analyzed in the T-Maze. At the decision point of the T-Maze, flies could choose for 2 min between the CS[+] and the CS[–] (OCT 1:670, MCH 1:1000). The performance index was calculated for MCH and OCT individually:

$$\text{Performance index} = (\# \text{ of flies}(CS^+) - \# \text{ of flies}(CS^-))/\text{total} \# \text{ of flies}$$
(2)

For each n the two data points obtained with MCH and OCT as CS+ were averaged.

## DopR function in l-LNv neurons of *D. melanogaster* adults

Flies were raised under 12 h:12 h light-dark cycles at 20 °C on standard fly food. One- to four-day-old male flies were placed individually in DAM (TriKinetics) monitors[83] containing 2% agar with 4% sucrose and 5 mM 9-*cis*-Retinal solved in ethanol (for opto-Dop1R1 and opto-Dop1R2) or only ethanol (for controls). The activity of the flies was recorded in complete darkness for 2 days before the flies were subjected to light pulses of 470 nm LED light with an intensity of 70 ± 10 µW/cm². The light pulses were administered 12 times during the previous light period of the 12 h:12 h light-dark cycle (one pulse every hour for 10 min, 15 min or 20 min). Experiments were performed 3 times with 32 experimental and control flies, respectively. Activity data were plotted as individual and average actograms using the ImageJ plug-in actogramJ[96], and individual and average activity profiles of the 24 h day with light pulses were calculated for each fly group as described in[97].

## Feeding behavior assays in *D. melanogaster* adults

Flies used in the flyPAD were reared and maintained in standard cornmeal food, with the composition described before[98] in incubators at 28 °C, 60% humidity and cycles of light/dark of 12 h each. After hatching, male flies of 4–8 days old were collected. Then, 5 µl of 10% sucrose solution containing 1% low gelling temperature agarose were placed in wells of the flyPAD containing electrodes to detect the capacitance change when the flies physically interacted with the food. The flies, following starvation for 24 h in the presence of a wet tissue with 3 ml of water, were transferred to the flyPAD individually using a pump. The experiments were all performed in a climate chamber at 25 °C, at 60% humidity. The recording of each session of flyPAD lasted 60 min, during which the flies could freely interact with the food.

For the optoPAD experiments[85], flies were reared and maintained in standard cornmeal food as explained above, with supplementation of all-*trans*-retinal at 0.2 mM concentration, in incubators at 25 °C, 60% humidity and blue light/dark cycles of 12/12 h. The chimeric dopaminergic receptors were activated using 523 nm green light at 3 V, which was automatically activated once the fly started to sip food. All flies were wet starved for 24 h prior to the experiment. The acquisition of the data was done using scripts (https://github.com/ribeiro-lab/optoPAD-software) based on Bonsai, an open-source program. The data analysis was done using Matlab (2022b).

## Statistics and reproducibility

No statistical method was used to predetermine the sample size. No data were excluded from the analyses except if samples did not meet sufficient quality standards, including sufficient cellular expression levels (HEK293 cell assays) or physically damaged samples after dissection. For functional imaging experiments, we excluded samples that showed significant z-drift during imaging. For analysis of larval locomotion, we excluded animals that could not be continuously tracked by the tracking software due to loss of signal. The experiments were randomized, and the investigators were blinded to allocation during experiments and outcome assessment whenever possible.

Statistical analysis was performed using Prism 8 (Graphpad, San Diego, CA, USA). All boxplots depict the median (center line) with 25th and 75th percentile (lower and upper box, respectively), and whiskers represent the 1st and 99th percentile. For line graphs, the mean ±SEM is shown. For high *n* numbers, violin plots with individual data points were used depicting the distribution of the data, including the 75th-percentile (upper dotted line), median (solid center line), and 25th-percentile (lower dotted line).

For the comparison of two groups, an unpaired two-tailed Student's *t*-test with Welch correction was used for normally distributed data, or, alternatively, a Mann-Whitney U-test for non-normally distributed data. A paired two-tailed Student's *t*-test was used for the comparison of the same individuals under different conditions (no light vs. light). One-way ANOVA with Dunnett's or Tukey's post hoc test was used for multiple comparisons. Statistical significance is defined as: *$p < 0.05$, **$p < 0.01$, ***$p < 0.001$.

Representative images were obtained from experiments that were repeated independently at least twice.

## Reporting summary

Further information on research design is available in the Nature Portfolio Reporting Summary linked to this article.

## Data availability

The raw data generated in this study are provided in the Source Data file. Due to the large size, raw imaging data (calcium imaging and immunohistochemistry) generated in this study can be obtained by request from the corresponding author. Requests will be fulfilled within 3 weeks. Source data are provided with this paper.

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

## Acknowledgements

We thank A. Thum for *H24-Gal4* and *MBON$^{g1/g2}$-Gal4*, M. Schwärzel for *UAS-bPAC*, J. Chu and Y. Li for *UAS-Gflamp1* fly lines. E. Kostenis and A. Inoue for HEK293ΔG7 cell lines. A. Schoofs for technical help with 2-Photon imaging. cDNA was obtained from the *Drosophila* Genomics Resource Center, supported by NIH grant 2P40OD010949. Stocks obtained from the Bloomington *Drosophila* Stock Center (NIH P40OD018537) were used in this study. This work was supported by grants from the Deutsche Forschungsgemeinschaft (DFG SO 1337/2-2 and SO 1337/7-1 to P.S., INST 248/293-1 to J.P., DFG FO207/14-1 to C.H.F., DFG GR-4310/11-1, DFG FOR2705 TP3, and INST 95/1419-1 to I.G.K.), the DFG Heisenberg program (SO1337/6-1 to P.S.). F.Z. was supported by a Chinese Scholarship Council (CSC) scholarship.

## Author contributions

F.Z. performed in vitro characterization and analysis of DopRs/opto-DopRs, cAMP signaling and larval behavior; A.M.T. and H.J. designed optoDopRs; B.N.I. performed and analyzed functional in vivo imaging; S.S. performed larval locomotion experiments and analysis; F.J.R.J., M.G.M., I.J. and I.G.K. designed, performed and analyzed flyPAD/opto-PAD experiments; M.H. and C.H.F. designed, performed and analyzed locomotor activity experiments; N.W., V.B. and J.P. designed, performed and analyzed adult learning and immunhistochemical experiments; T.L. and K.S. contributed to in vitro characterization of DopRs/optoDopRs; P.S. designed the study and wrote the manuscript with input from all authors.

## Funding

## Competing interests

The authors declare no competing interests.
