## [Peer Review File · Nature Communications]

Reviewers' comments:

Reviewer #1 (Remarks to the Author)

The manuscript „Optimized design and in vivo application of optogenetically functionalized *Drosophila* dopamine receptors” describes new optogenetic approach, which may be very useful in dopaminergic studies and may be helpful in the designing of other optogenetically regulated receptors. In general, paper is very nicely prepared, starting from detailed description of optoDopR constructs, experiments in vitro checking whether they are working, ending with in vivo experiments giving results in behavioral assays. Figures are very good quality and table S1 is very informative. I strongly recommend to publish this paper.

I have just minor comments:

1. In the introduction Authors could mention about differential role of dopamine in specific brain area, which causes many side effects during Parkinson, ADHD and Schizophrenia treatment based on systemic dopaminergic pathway modulation. This information would explain a reason why to study single dopamine receptors function in details.
2. Fig 1C appears in the text before Fig1B which is a little bit confusing
3. More detailed description of figure legend, like what is RLU on the Y axe. Fig 3 – what is visualized as GFP, S4 – what is AM? Description is only in the main text but not in figure legend
4. Please explain the reason of retinal feeding – I think not all readers will be familiar with this technique
5. PDF should be written with big letters as a neuropeptide, Pdf-Gal4 strain is not specific for l-LNv only, but also for s-LNv
6. Please explain in methods why you are using anti-Rhodopsin immunostaining, or in the main text/figure legends that optoDopR were stained as Rh.
7. Please explain why you are using anti-Dlg1 staining, there is also no dilution/source in the methods
8. Figure legend S2 and S3 are reversed: S2 describes S3 and legend S3 describes S2 – please correct
9. Figure S3D (in the legend S2D) does not show light-intensity dependent signaling for optoDop1R2v2, but rather G protein coupling properties of optoDop1R1V2 after activation with light
10. Why innate preference for 3-octanol in control is not aversive?

Reviewer #2 (Remarks to the Author)

In this manuscript, the authors establish two optogenetically inducible dopamine receptor (optoDopR) and functionally evaluated the receptor in vivo in *Drosophila*. They improved the design of the chimera and show that leaving the first intracellular loop of the rhodopsin intact and adapting the C-terminus tail, results in functional chimeras for the Dop1R1 and Dop1R2. Whereas the initial evaluation is in cell lines, they expressed the construct in *Drosophila* and studied the consequences of turning the chimera

on the behavioral response of drosophila larvae and adult *Drosophila*. The authors identified a cell type-specific in vivo response.

Overall, the study provides exciting new insights and proves the functionality of the receptors also in vivo setting providing a strong case. Since the manuscript is for a broad audience, the explanation of the results should be explained to a broader audience since groups designing chimeras will likely not have deep insights into the *drosophila* morphology, the dopamine receptor response, and behavior and vice versa. I am providing below a couple of examples that were less straightforward to follow. An additional challenge in evaluating the manuscript is the mix up of supplementary Figures 1-3 and their figure legends (see below) causing further confusion. The authors have to work this out carefully in their revised manuscript.

Examples of required clarifications:

L53- : The behavior, to GPCR to Dopamine is not obvious because the key message is missing that dopamine receptors are GPCR.

L62-L64: Clarification needed why pharmacological approaches miss precision?

L96-: Explanation needed for why the authors focus only on D1-receptor, and not also on Dop2R and DopEcR? How does the D1-receptor relate to certain behaviors that will be later evaluated? It would be crucial to explain how Dop1R1 and R2 differ in their G-protein signaling. Only in L226-229, the expected G proteins are adequately explained, which makes it challenging to follow the results before.

L108: In which way were the subcellular localization “strongly improved”?

L128- : The authors mentioned “six other GPCRs”? To which GPCRs, they are referring to.

L129-: How have the authors decided that they fail to produce functional chimeras? The authors refer here to Fig. 1C and data not shown but at this point it is not clear what is Fig. 1C showing and why there are several different G-proteins shown.

L294- : “as previously described”. Are the authors referring to an experiment, or to a reference?

Fig. 1-3/Supp. Fig. 1-3:

The experimental description to validate the candidates are only superficially described for Fig. 1B and onwards. It would be beneficial to show a schematic of the GloSensor cAMP reporter assay to explain the strategy.

Figure and Figure legend mix-up: Suddenly in Fig. S1b results for Gi and Go are shown but the text talks about Gs and G15, which causes several confusions in reading the manuscript. I am assuming that Fig. S1 shows the results of Dop1R2, whereas Fig. S3 shows Dop1R1 but has the figure legend form Figure S2.

A challenge in the GPCR field is to have the appropriate endogenous receptor localization (L479-480). The description of the experiment and the result have been too superficial e.g. it does not come across how the optoDop1R1/R2 transgene is generated and how the expression of the optoDop1R1/R2 has been confirmed (Rhodopsin antibody?). Was the expression mostly on the cell surface, or did the optoDop1R1/R2 also accumulated within the cell body? Higher resolution images on a single cell level would be beneficial to confirm the surface expression similar as in Fig. S2a (Fig. legend S3a?).

Do the authors have analyzed whether the optoDop1R1/R2 expression overlapped with the original Dop1R1/R2 expression in larvae and adult animals?

For the functionality analysis, the authors should focus in describing their rationale for R1 and R2 knockout and what is known specifically about these receptors. Specifically, R2 comes up rather sudden in L329 without further explanation. To clarify these aspects will significantly improve the manuscript.

Fig. 5g-l, 6C-F show a high error bar and are missing the dots for the individual samples. For Fig. 5g-l, it would be beneficial to show the connection between the individual time points, since they are coming from the same animal?

“Data not shown” should be only used in very rare location, especially in light of the possibility of supplementary Figures. L254 experiments would be important to show.

It has been not clear how the “Rolling” in Figure 3D is quantified. It is just one time rolling or a twitching? If an animal rolled repeatedly is this counted as several time?

Reviewer #3 (Remarks to the Author)

The chimeric optoXR approach was utilized by the authors to optimize highly functional and specific optoDopRs. These tools enabled in vivo analysis of two of these receptors' specific function in *Drosophila* behavior (locomotion and learning processes). Their findings describe the sensitivity, proper cellular distribution and G-protein engagement induced by their activation in vitro and in vivo in specific set ups. Importantly, they demonstrated that only optoDop1R1V2 activation, and not optoDop1R2V2, promoted LNV-mediated arousal. Conversely, operant activation of optoDop1R2V2, but not optoDop1R1V2, in valence-encoding MBONs controlled feeding.

As already mentioned, the authors generated and optimized chimeric optoXRs of *Drosophila melanogaster* Dop1R1 and Dop1R2 by exploiting the evolutionary constraints of G protein coupling specificity. The generation of these optogenetic tools was correctly justified by the importance of spatiotemporal manipulation of the GPCR (particularly modulatory neurotransmitters) function. However, the complementation of DopR KO animals by activating the optoDopR performed by the authors, indicates that kinetics is not a major factor in this context. Additionally, no kinetic studies of the transduction proteins or comparative second messenger kinetics were performed in the present work.

The chimeric G α s protein ('Gsx') assay was utilized to directly compare the coupling and kinetics of GPCRs with the major G-proteins upstream of the cAMP reporter GloSensor.

Figure 1 should include statistical analysis and significance, as observed in other figures.

Supplementary Figure S1A describes the kinetic activation of Dop1R2. However, in the results section (line 144) and in its legend, it refers to Dop1R1. This is not trivial since Dop1R2 should activate Gq preferentially, as previously described (Himmelreich et al., 2017). Moreover, the activation of Gq by Dop1R2 is not preferentially activated with the optoDop1R2, as presented in Figure S1C. This is especially confusing since a similar strategy using the 'Gsx' assay and the same cell type (HEK293T cells) was used in Himmelreich et al., showing the preferential activation of Gq by Dop1R2. This should be at least mentioned and discussed.

In line 187, the authors claim, "Similarly to the wildtype receptor, optoDop1R2v2 coupled to the same G proteins, prominently with Gs and G15 showing light dose-dependent responses...". As already mentioned, Dop1R2 should primarily activate Gq signaling, consistent with its effect on calcium release. In line 195, the authors state that their strategy to design optogenetically activatable GCPR receptors allows specificity. However, as previously mentioned, optoDop1R2v2 does not activate Gq as it should.

Later, in line 227, the authors mention, "Dop1R1 has been reported to be primarily linked to Gs-dependent cAMP production, while Dop1R2 can induce intracellular calcium release via activation of Gq-family signaling that includes G15." In fact, there is activation of G15 by Dop1R2 and optoDop1R2v2, but the proportion of activation compared with Gs is inverted compared to the results presented by Himmelreich et al. 2017.

The inability of optoDop1R2 to activate Gq in vitro contrasts with its capability to rescue the phenotype of Dop1R2 KO. This is equally intriguing and should be mentioned, providing some explanation in the discussion section. In the legend of FigS1, there are references to optoDop1R1v2 in panels C, D and E, but the corresponding graphs are labeled as optoDop1R2v2 in each case.

Regarding the characterization of Dop1R2, in line 184 the authors claim, "Dop1R2 showed dose-dependent coupling to Gs, G15, and inhibitory G proteins upon addition of dopamine in the range of 0.1-100 nM (Fig. 2B, C, S2A, B)..." However, in Figure 2C, it is evident that the dose dependency occurs in the range of 0.1-5 nM, as the responses become constant at higher concentrations, reaching a plateau. Additionally, the reference to figure S2 in this case is incorrect. Figures S1 and S2, along with their legends, are incorrect and should be revised. The same issue occurs with Figure S3, where the figure itself appears to correspond to the legend presented in Figure S2.

In the results section, when first presented, the authors should mention the kind of transgenes they generated and how they induce their expression. This is particularly important to understand the fact that both optoXR are induced/transcribed at equal levels, and variations in protein levels likely result from variations in stability and/or distribution. In this regard, the analysis of their distribution should be compared not only between the optoDopR1 and 2, and their versions, but also importantly in relation to their endogenous distributions. The location of the soma, axon, and calyx of the Kenyon Cells should be showed in the confocal images. This would help in understanding the observed differences in distribution.

Since the in vitro studies presented in the manuscript indicate that both optoDop1R1 and optoDop1R2 activate Gs at higher levels, and both activate G15 to a minor extent, but similarly, in fact the ratio of Gs/G15 activation appears to be similar in both cases, it would be important to compare the levels of cAMP and calcium induced by both optoDop1Rs in KC. This is not presented and authors only show the expected results, Dop1R1 increasing cAMP and Dop1R2 increasing intracellular calcium.

The recovery of dopaminergic signaling/function (reward learning) in KD animals (Dop1R1 IR) or in rotenone-treated animals (locomotion) should also be performed using the optoDop1R2v2 construct, considering that both receptors are expressed in the same cells, as shown by single-cell transcriptomics and endogenous GFP-tagged dop receptors.

It is remarkable that, as the authors claim in line 312, "Using optoDop1R1V1 or optoDop1R1V2 expression in KCs under these conditions partially rescued fructose-odor learning (Fig. S4B, C)...", this although V1 and V2 versions possess differences in levels of expression and cellular distribution. Thus, for in vivo function in the reward learning test at least, the variation in levels, specificity, and distribution appears to be unimportant. In fact, it appears that Dop1R1V1 better recovers the learning performance. This paradox should be stressed in other assays presented in this work and further reinforce the importance of their chimeric Rho-GPCR design.

In Figure 5, the authors show how activation of Dop1R1 boosts the activity during the first hours of the circadian cycle and indicate that "...we also tested optoDop1R2V2 activation under the same conditions but did not observe a significant effect on blue-light-induced activity (Fig. 5I, S5A-C)..." and conclude "These findings suggest a specific role for Dop1R1 signaling in I-LNvs promoting morning activity upon arousal." It would be interesting to test, using G protein-specific RNAi, whether the differential requirement is due to the levels of expression, cellular distribution, or G protein engaged by the optoDop1R2 construct. Is Dop1R1V1 also able to produce this effect on blue-light-induced activity?

It would be helpful to explain why sometimes the activation of the optoDop receptors is performed with blue light and in other cases with green light.

Finally, in the discussion section, the authors claim that spectral differences between optoDop1R constructs and other light-induced proteins (like Crimson, for instance) would allow their combined activation and directly address the modulatory influence of dopamine on the activation parameters of specific neurons. This is fascinating, and it would be more attractive if the authors could propose a specific situation in which it could be implemented, such as epileptic conditions or learning processes, for example.

In line 353, a closing parenthesis is missing: "MBON-g1/g2 (adapted from 50.)"

Reference 71 is incorrectly written.

These are some critical points and concerns regarding the manuscript. It would be beneficial for the authors to address these issues and make the necessary revisions before considering for publication.

Response to the reviewers

We thank all reviewers for their constructive comments and suggestions strongly improving our manuscript. Please find detailed responses to the individual points below.

Reviewer #1 (Remarks to the Author)

The manuscript „Optimized design and in vivo application of optogenetically functionalized Drosophila dopamine receptors” describes new optogenetic approach, which may be very useful in dopaminergic studies and may be helpful in the designing of other optogenetically regulated receptors. In general, paper is very nicely prepared, starting from detailed description of optoDopR constructs, experiments in vitro checking whether they are working, ending with in vivo experiments giving results in behavioral assays. Figures are very good quality and table S1 is very informative. I strongly recommend to publish this paper.

I have just minor comments:

Response: we thank the reviewer for the very positive assessment of our work, and we have addressed all minor concerns mentioned below.

1. In the introduction Authors could mention about differential role of dopamine in specific brain area, which causes many side effects during Parkinson, ADHD and Schizophrenia treatment based on systemic dopaminergic pathway modulation. This information would explain a reason why to study single dopamine receptors function in details.

Response: we thank the reviewer for this suggestion, and we have implemented a short paragraph on these aspects in the introduction (see l66 ff).

2. Fig 1C appears in the text before Fig1B which is a little bit confusing

Response: we have amended this issue and rearranged the text accordingly.

3. More detailed description of figure legend, like what is RLU on the Y axe. Fig 3 – what is visualized as GFP, S4 – what is AM? Description is only in the main text but not in figure legend.

Response: we have now added the missing information explaining RLU (relative light units), the specifics of the proteins visualized and other abbreviations including AM in the respective figure legends.

4. Please explain the reason of retinal feeding – I think not all readers will be familiar with this technique.

Response: we have now included an explanation for retinal feeding in the respective section when we describe the in vivo experiments (see p14, l325 ff).

5. PDF should be written with big letters as a neuropeptide, Pdf-Gal4 strain is not specific for I-LNv only, but also for s-LNv.

Response: the reviewer is correct, and we have revised our statements in the respective section accordingly (p24, l542ff).

6. Please explain in methods why you are using anti-Rhodopsin immunostaining, or in the main text/figure legends that optoDopR were stained as Rh.

Response: we thank the reviewer for noticing this point. The optoXR constructs contain a C-terminal Rhodopsin sequence (TETSQVAPA) with the 1D4 anti-Rhodopsin epitope. We have now added the missing information in the respective methods and included the sequence information of all generated constructs in Table S2.

7. Please explain why you are using anti-Dlg1 staining, there is also no dilution/source in the methods

Response: we have now explained the use of anti-Dlg in the respective figure legends (Fig. 3 and Fig. S4) and included the details (dilution, source) in the respective methods section.

8. Figure legend S2 and S3 are reversed: S2 describes S3 and legend S3 describes S2 – please correct

Response: we apologize for the mix-up and have now corrected the respective figure order and legends.

9. Figure S3D (in the legend S2D) does not show light-intensity dependent signaling for optoDop1R2v2, but rather G protein coupling properties of optoDop1R1V2 after activation with light

Response: we thank the reviewer for pointing out this mix-up due to the wrong supplementary figure assignment. We have now corrected the respective figure order and legends.

10. Why innate preference for 3-octanol in control is not aversive?

Response: we are not aware of the stated aversive nature of 3-octanol. 3-octanol has been used for aversive conditioning (e.g. (Aceves-Piña & Quinn, 1979)). Based on the literature we found, 3-octanol is attractive to naïve larvae², which is consistent with our results.

Reviewer #2 (Remarks to the Author)

In this manuscript, the authors establish two optogenetically inducible dopamine receptor (optoDopR) and functionally evaluated the receptor in vivo in *Drosophila*. They improved the design of the chimera and show that leaving the first intracellular loop of the rhodopsin intact and adapting the C-terminus tail, results in functional chimeras for the Dop1R1 and Dop1R2. Whereas the initial evaluation is in cell lines, they expressed the construct in *Drosophila* and studied the consequences of turning the chimera on the behavioral response of *Drosophila* larvae and adult *Drosophila*. The authors identified a cell type-specific in vivo response.

Overall, the study provides exciting new insights and proves the functionality of the receptors also in an in vivo setting providing a strong case. Since the manuscript is for a broad audience, the explanation of the results should be explained to a broader audience since groups designing chimeras will likely not have deep insights into the *Drosophila* morphology, the dopamine receptor response, and behavior and vice versa. I am providing below a couple of examples that were less straightforward to follow. An additional challenge in evaluating the manuscript is the mix up of

supplementary Figures 1-3 and their figure legends (see below) causing further confusion. The authors have to work this out carefully in their revised manuscript.

Response: We thank the reviewer for the positive evaluation of our work and the insightful comments. We have carefully revised the manuscript according to the reviewer's suggestions, amended missing information and resolved the supplementary figure mix-up.

Examples of required clarifications:

L53- : The behavior, to GPCR to Dopamine is not obvious because the key message is missing that dopamine receptors are GPCR.

Response: We thank the reviewer for pointing this out and we have now clarified this point in our introduction (p3 l58 ff).

L62-L64: Clarification needed why pharmacological approaches miss precision?

Response: We have revised our original statement and now state that "pharmacological approaches are not cell type specific and difficult to control temporally, thus lacking the precision and specificity to target defined circuits and their regulated behaviors".

L96-: Explanation needed for why the authors focus only on D1-receptor, and not also on Dop2R and DopEcR? How does the D1-receptor relate to certain behaviors that will be later evaluated? It would be crucial to explain how Dop1R1 and R2 differ in their G-protein signaling. Only in L226-229, the expected G proteins are adequately explained, which makes it challenging to follow the results before.

Response: We have now introduced D1/D2-receptor type signaling in the introduction (p3 l61ff), as well as the differences in signaling of Dop1R1 and Dop1R2 (p5 l107 ff) to clarify their functions early on.

We did not focus on Dop2R and DopEcR for several reasons: first, the neuronal substrates and actions of Dop1Rs are better studied and were more accessible for our proof of principle analysis. Secondly, for Dop2R, we had difficulties measuring the activity of the native receptor in our assays and the multiple reported isoforms made it more complex to design respective opto-receptors. Thus, it will be highly interesting to tackle Dop2R and also DopEcR opto-designs in future studies.

L108: In which way were the subcellular localization "strongly improved"?

Response: we have now stated that axonal and dendritic localization of the V2 optoDopRs was strongly improved (p5 l123).

L128- : The authors mentioned "six other GPCRs"? To which GPCRs, they are referring to.

Response: We have now included the other tested GPCRs designed with the original V1 approach in the new supplementary Figure 1 (Fig. S1 and Table S2).

L129-: How have the authors decided that they fail to produce functional chimeras? The authors refer here to Fig. 1C and data not shown but at this point it is not clear what is Fig. 1C showing and why there are several different G-proteins shown.

Response: We have tested all produced chimeras in the Gsx assay. In case no significant response was observed (compared to control cells that were not transfected with a Gsx variant), we call this particular construct non-functional. We have now included a schematic and more comprehensive introduction of the Gsx assay before showing any data (p6 l144 ff, and Fig. 1b). We now also included all Gsx assay data on the V1 optoXRs we have tested (see Fig. 1c, S1, Table S2).

L294- : “as previously described”. Are the authors referring to an experiment, or to a reference?

Response: we were referring to a reference which was not positioned correctly. We have now amended this issue (p18 l418).

Fig. 1-3/Supp. Fig. 1-3:

The experimental description to validate the candidates are only superficial described for Fig. 1B and onwards. It would be beneficial to show a schematic of the GloSensor cAMP reporter assay to explain the strategy.

Response: We thank the reviewer for this suggestion, and we have added a schematic and more explanation of the GloSensor assay (p6 l144 ff and Fig. 1b). We have also included a schematic for the TRUPATH assay shown in the supplementary figures (Fig. S2f).

Figure and Figure legend mix-up: Suddenly in Fig. S1b results for Gi and Go are shown but the text talks about Gs and G15, which causes several confusions in reading the manuscript. I am assuming that Fig. S1 shows the results of Dop1R2, whereas Fig. S3 shows Dop1R1 but has the figure legend form Figure S2.

Response: We apologize for the figure mix-up, which we have now cleared up and arranged all supplementary figures in the appropriate order with corresponding legends.

A challenge in the GPCR field is to have the appropriate endogenous receptor localization (L479-480). The description of the experiment and the result have been too superficial e.g. it does not come across how the optoDop1R1/R2 transgene is generated and how the expression of the optoDop1R1/R2 has been confirmed (Rhodopsin antibody?). Was the expression mostly on the cell surface, or did the optoDop1R1/R2 also accumulated within the cell body? Higher resolution images on a single cell level would be beneficial to confirm the surface expression similar as in Fig. S2a (Fig. legend S3a?).

Response: We thank the reviewer for these comments, and we have now more clearly described the transgenesis approach and the immunostaining procedure to localize optoDopRs, for which we indeed used a Rhodopsin antibody directed against the C-terminal Rho sequence attached to the C-terminus of our optoXRs (see Table S2).

Importantly, we have added additional analyses on the localization of optoDopRs including quantitative data on their subcellular distribution (see Fig. 3c). Moreover, we have now compared optoDopR localization at the single cell level in larval (MBONs) and adult Kenyon cells showing the differences between the V1 and V2 design in expression and localization

(see Fig 3f,h, S4b,e,f). While it is difficult to prove surface expression in vivo, we did observe accumulation of optoDopRs along dendrites and axonal terminals together with a membrane-bound marker (CD4-tdTomato or myr-tdTomato) suggesting at least some surface localization.

Do the authors have analyzed whether the optoDop1R1/R2 expression overlapped with the original Dop1R1/R2 expression in larvae and adult animals?

Response: We have now added data on the endogenous localization of Dop1R1 in relation to our optoDop1R1. Due to the identical insertion site of some of the transgenes we were only able to do side by side comparison, which we however did extensively down to the single cell level. We have quantified the relative distribution of the endogenous Dop1R1 and optoDop1R1 in larval Kenyon cells (Fig. 3d,e). We further show optoDopR localization in single MBONs (Fig. 3f, S4b) and adult Kenyon cells, showing that optoDop1R1^{V2} but not V1 displays a similar subcellular distribution as endogenous Dop1R1.

For Dop1R2 no knock-in line allowing labeling of endogenous receptor is currently available. Based on personal communication with the authors who generated the Dop1R1-GFP11 knock-in line³, this approach did not work for Dop1R2 as it resulted in a mislocalized receptor (R. Tanimoto, personal communication). As no suitable antibodies could be obtained, we focused our analyses on Dop1R1, for which we showed comparable localization between the wildtype and opto-receptor.

For the functionality analysis, the authors should focus in describing their rationale for R1 and R2 knockout and what is known specifically about these receptors. Specifically, R2 comes up rather sudden in L329 without further explanation. To clarify these aspects will significantly improve the manuscript.

Response: We thank the reviewer for this suggestion. We have now introduced the Dop1R1/R2 receptors more clearly in our introduction (p5 l107 ff) and explicitly describe their known functions there. We further mention specific aspects of their function in the relevant sections where we tested receptor dependent behaviors in knockout animals, with the goal to exemplify their functions more clearly (p18 l414, p20 l461f).

Fig. 5g-I, 6C-F show a high error bar and are missing the dots for the individual samples. For Fig. 5g-I, it would be beneficial to show the connection between the individual time points, since they are coming from the same animal?

Response: We originally omitted the individual data points due to the high n numbers in these panels. We therefore now plotted the respective data as violin plots (new Fig. 6g-i, Fig 7c-f) showing all data points and displaying the distribution of the data. For Fig.5g-i (new Fig. 6g-i) we could not depict the connections of the individual animals as we are comparing two independent datasets, with and without retinal feeding, to show the specific effect of optoDopR activation. We have however now included paired analyses of the individuals from one group showing blue light-induced activity increase for optoDop1R1 but not optoDop1R2 groups (see Fig.S7a,d)

“Data not shown” should be only used in very rare location, especially in light of the possibility of supplementary Figures. L254 experiments would be important to show.

Response: We agree with the reviewer and have now included respective data for repeated activation of optoDop1R1 and optoDop1R2 showing consistent induction of cAMP increase and calcium transients, respectively (Fig. 4k,l).

It has been not clear how the “Rolling” in Figure 3D is quantified. It is just one time rolling or a twitching? If an animal rolled repeatedly is this counted as several time?

Response: We have now clarified the analysis protocol in the corresponding methods part. Each animal was only counted once, and the strongest behavioral response was considered. Rolling is defined as a full 360° turn along the larval body axis, and we did not differentiate between single or multiple rolling, as the main purpose of this assay was to get an initial assessment of optoDopR functionality.

Reviewer #3 (Remarks to the Author)

The chimeric optoXR approach was utilized by the authors to optimize highly functional and specific optoDopRs. These tools enabled in vivo analysis of two of these receptors’ specific function in Drosophila behavior (locomotion and learning processes). Their findings describe the sensitivity, proper cellular distribution and G-protein engagement induced by their activation in vitro and in vivo in specific set ups. Importantly, they demonstrated that only optoDop1R1V2 activation, and not optoDop1R2V2, promoted LNV-mediated arousal. Conversely, operant activation of optoDop1R2V2, but not optoDop1R1V2, in valence-encoding MBONs controlled feeding.

As already mentioned, the authors generated and optimized chimeric optoXRs of Drosophila melanogaster Dop1R1 and Dop1R2 by exploiting the evolutionary constraints of G protein coupling specificity. The generation of these optogenetic tools was correctly justified by the importance of spatiotemporal manipulation of the GPCR (particularly modulatory neurotransmitters) function. However, the complementation of DopR KO animals by activating the optoDopR performed by the authors, indicates that kinetics is not a major factor in this context. Additionally, no kinetic studies of the transduction proteins or comparative second messenger kinetics were performed in the present work.

Response: We thank the reviewer for the insightful comments. We agree that we did not perform kinetic studies in our work, thus we also could not draw conclusions about the kinetics of receptor signaling and function. We would however not claim that kinetics do not matter as our rescue experiments in some cases did not fully recover behavioral function (e.g. larval learning assays). As our work represents the first characterization of *Drosophila* optoXRs, we focused on proof of principle experiments to show the utility of these novel tools, leaving room for future more detailed studies on these aspects.

The chimeric G_s protein ('G_{sx}') assay was utilized to directly compare the coupling and kinetics of GPCRs with the major G-proteins upstream of the cAMP reporter GloSensor. Figure 1 should include statistical analysis and significance, as observed in other figures.

Response: We have added the statistical information in the panels of Fig. 1 and 2. We would like to add that the kinetics of the G_{sx} assay are slow (see below) and we therefore did not perform any analysis on it, as it does not represent the receptor-G protein dissociation kinetics.

Supplementary Figure S1A describes the kinetic activation of Dop1R2. However, in the results section (line 144) and in its legend, it refers to Dop1R1. This is not trivial since Dop1R2 should activate G_q preferentially, as previously described (Himmelreich et al., 2017). Moreover, the activation of G_q by Dop1R2 is not preferentially activated with the optoDop1R2, as presented in Figure S1C. This is especially confusing since a similar strategy using the 'G_{sx}' assay and the same cell type (HEK293T cells) was used in Himmelreich et al., showing the preferential activation of G_q by Dop1R2. This should be at least mentioned and discussed.

Response: We sincerely apologize for our mistake of not properly arranging the supplementary figures. We have amended this issue and the figures and legends are now in the appropriate order and new Figure S2 (formerly S1) now describes the kinetics of Dop1R1 activation.

In line 187, the authors claim, "Similarly to the wildtype receptor, optoDop1R2V2 coupled to the same G proteins, prominently with G_s and G₁₅ showing light dose-dependent responses...". As already mentioned, Dop1R2 should primarily activate G_q signaling, consistent with its effect on calcium release. In line 195, the authors state that their strategy to design optogenetically activatable GPCR receptors allows specificity. However, as previously mentioned, optoDop1R2v2 does not activate G_q as it should.

Later, in line 227, the authors mention, "Dop1R1 has been reported to be primarily linked to G_s-dependent cAMP production, while Dop1R2 can induce intracellular calcium release via activation of G_q-family signaling that includes G₁₅." In fact, there is activation of G₁₅ by Dop1R2 and optoDop1R2v2, but the proportion of activation compared with G_s is inverted compared to the results presented by Himmelreich et al. 2017. The inability of optoDop1R2 to activate G_q in vitro contrasts with its capability to rescue the phenotype of Dop1R2 KO. This is equally intriguing and should be mentioned, providing some explanation in the discussion section.

Response: We agree with the reviewer that the data of our G_{sx} assay does not fully reflect the previously reported activity of Dop1R2. The G_{sx} employed by us, and G protein fingerprinting assay employed by Himmelreich et al. have different abilities to detect G protein coupling and activity. The G_{sx} assay employs chimeric G proteins which contain the GPCR binding region of specific G_x proteins (G_i, o, t, z, q, 12, 13, or 15) linked to the G_s signaling domain, which induces adenylate cyclase activity⁴. Thus, the downstream readout for G_{sx} coupling is highly comparable and quantitative, but kinetics are slow due to the rate limiting activity of the employed cAMP-dependent luciferase reporter (Glo22F). G protein fingerprinting is based on the dissociation of G_{βγ} from G_α upon GPCR activation, which results in BRET signal increase by Venus-tagged G_{βγ} interaction with Nanoluc-tagged GRK⁵. The G_{sx} assay and fingerprinting both allow to probe the G protein space but fingerprinting additionally enables the analysis of dissociation kinetics. However, fingerprinting is not easily

implemented with optoXRs and less robust compared to the G_{sx} assay, which is why we originally implement this assay. We now added data from a recently developed TRUPATH assay which also allows kinetic measurement of G protein coupling based on $G_{\alpha}G_{\beta/\gamma}$ dissociation ⁶. Using this assay, we obtained similar coupling profiles as in our G_{sx} assay for Dop1R and optoDop1R1 (Fig. S2e-g). For Dop1R2 and optoDop1R2 we mostly observed G15 activation suggesting that the dissociation kinetics is much faster for G15.

It is possible that the chimeric Gq proteins employed in the G_{sx} and TRUPATH assays might not be able to bind Dop1R2 efficiently thus not properly reporting its activity. However, we would like to emphasize that G15 belongs to the Gq family and also induces signaling via intracellular calcium store release ⁷. While these assays allow sampling the possible G protein partners to some extent, what happens *in vivo* always depends on the cell type and the expressed G protein subsets.

Consistently, we added additional data showing that opto-Dop1R2v2 preferentially induces calcium release, while optoDop1R1 preferentially signals via cAMP in *Drosophila* mushroom body neurons *in vivo* (see Fig. 4, S5).

In the legend of FigS1, there are references to optoDop1R1v2 in panels C, D and E, but the corresponding graphs are labeled as optoDop1R2v2 in each case.

Response: We sincerely apologize for our mistake of not properly arranging the supplementary figures. We have amended this issue and the figures and legends are now in the appropriate order and describe the respective panels correctly.

Regarding the characterization of Dop1R2, in line 184 the authors claim, "Dop1R2 showed dose-dependent coupling to Gs, G15, and inhibitory G proteins upon addition of dopamine in the range of 0.1-100 nM (Fig. 2B, C, S2A, B)..." However, in Figure 2C, it is evident that the dose dependency occurs in the range of 0.1-5 nM, as the responses become constant at higher concentrations, reaching a plateau. Additionally, the reference to figure S2 in this case is incorrect. Figures S1 and S2, along with their legends, are incorrect and should be revised. The same issue occurs with Figure S3, where the figure itself appears to correspond to the legend presented in Figure S2.

Response: We have now correctly organized and described all the panels. We also fixed the wording regarding the range of dose-dependency, which was correctly pointed out by the reviewer.

In the results section, when first presented, the authors should mention the kind of transgenes they generated and how they induce their expression. This is particularly important to understand the fact that both optoXR are induced/transcribed at equal levels, and variations in protein levels likely result from variations in stability and/or distribution. In this regard, the analysis of their distribution should be compared not only between the optoDopR1 and 2, and their versions, but also importantly in relation to their endogenous distributions. The location of the soma, axon, and calyx of the Kenyon Cells should be showed in the confocal images. This would help in understanding the observed differences in distribution.

Response: we agree with the reviewer's points and have now introduced the transgenic approach more clearly, stating that we used the same genomic integration site which allows for comparable expression (p11 I251f). In addition, we have now clearly labeled the mushroom body compartments and extended the analysis of DopR localization (see new Fig.

3). We now show the differential distribution of the V1 and V2 optoDopRs quantitatively (see Fig. 3 a-c).

Importantly, we have added data on the endogenous localization of Dop1R1 in relation to our optoDop1R1. Due to the identical insertion site of some transgenes we were only able to do side by side comparison, which we however did extensively down to the single cell level. We have quantified the relative distribution of the endogenous Dop1R1 and optoDop1R1 in Kenyon cells showing their similar subcellular distribution (Fig. 3d,e). In addition, we now show single cell localization of endogenous Dop1R1 and our optoDopRs in larval MBONs (Fig. 3f, S4b) and adult Kenyon cells (Fig. 3g,h, S4e,f). Based on these analyses we conclude that optoDop1R1^{V2} and endogenous Dop1R1 have a comparable subcellular localization in dendrites and axons, while optoDop1R1^{V1} is more weakly expressed and mostly accumulates in the cell soma.

As there are no suitable tools for investigating endogenous Dop1R2 localization we cannot draw conclusions about the similarity in distribution with optoDop1R2^{V2}. However, functional data on Dop1R2 contribution to calcium release along Kenyon cell axons⁸ supports its presence along these structures, which is consistent with the localization of optoDop1R2^{V2}.

Since the *in vitro* studies presented in the manuscript indicate that both optoDop1R1 and optoDop1R2 activate Gs at higher levels, and both activate G15 to a minor extent, but similarly, in fact the ratio of Gs/G15 activation appears to be similar in both cases, it would be important to compare the levels of cAMP and calcium induced by both optoDop1Rs in KC. This is not presented and authors only show the expected results, Dop1R1 increasing cAMP and Dop1R2 increasing intracellular calcium.

Response: We agree with the reviewer and we have added extensive data to the manuscript, which provides important information about the specificity of the generated optoDopRs. We now show that optoDop1R1^{V2} preferentially activates cAMP signaling with virtually no effect on calcium release in Kenyon cells (see Fig. 4e-j, S5d,h,i). Conversely, optoDop1R2^{V2} shows the expected specificity for calcium signaling and only minor cAMP response induction (see Fig. 4e-j, S5e-g), confirming that the differential signaling properties of Dop1R1 and Dop1R2 are preserved in our optoDopRs. We now also show that optoDop1R1^{V1} activation results in weak cAMP induction mostly in the soma of Kenyon cells (Fig. S5a-c).

We would like to point out that the *in vitro* results in cells only show the G protein coupling profile of a receptor, which are frequently promiscuous in regard to G protein specificity. The extent to which these receptors use the possible pathways *in vivo* is however determined by its localization, the expressed G protein repertoire and additional cofactors (e.g. see⁵). This notion is now also evident from our comprehensive *in vivo* data showing that the optoDopRs show preferential signaling via the previously described pathways as well as cell type specific functions.

The recovery of dopaminergic signaling/function(reward learning) in KD animals (Dop1R1 IR) or in rotenone-treated animals (locomotion) should also be performed using the optoDop1R2v2 construct, considering that both receptors are expressed in the same cells, as shown by single-cell transcriptomics and endogenous GFP-tagged dop receptors.

Response: As suggested by the reviewer, we performed the locomotion experiment with rotenone treatment using the same Dop1R1 knock-in Gal4 driver line expressing optoDop1R1^{V1} or optoDop1R2^{V2}. First, we now show that optoDop1R1^{V1} cannot rescue rotenone-induced locomotion defects (Fig. 5b, S6a). Secondly, optoDop1R2^{V2} activation can

partially but not fully rescue larval velocity but not turning behavior (Fig. 5d, S6d). In contrast, optoDop1R1^{V2} shows the strongest rescue activity and we now also show that this is specific for rotenone-treated animals, as activation of optoDop1R1^{V2} in non-treated animals has no effect on locomotion (Fig. S6c). This data thus suggests that optoDop1R1^{V2} induced signaling, and to some extent also optoDop1R2, can functionally recover locomotion defects induced by the loss of DAergic neuron function.

It is remarkable that, as the authors claim in line 312, "Using optoDop1R1V1 or optoDop1R1V2 expression in KCs under these conditions partially rescued fructose-odor learning (Fig. S4B, C)...", this although V1 and V2 versions possess differences in levels of expression and cellular distribution. Thus, for in vivo function in the reward learning test at least, the variation in levels, specificity, and distribution appears to be unimportant. In fact, it appears that Dop1R1V1 better recovers the learning performance. This paradox should be stressed in other assays presented in this work and further reinforce the importance of their chimeric Rho-GPCR design.

Response: we respectfully disagree that there is no improvement of optoDop1R1^{V1} in relation to V2. First, expression as well as specificity of the V2 design are significantly improved, additionally yielding optoDop1R2^{V2} which was nonfunctional as V1. Second, localization and signaling function *in vivo* are also strongly improved compared to V1. While both optoDop1R1^{V1} and optoDop1R1^{V2} can partially rescue learning behavior in larvae to a similar extent (the differences between their rescue activity are not statistically significant), we do not think it can be concluded that localization and signaling specificity do not matter. The difficulty of eliciting compartment specific activation of optoDopR in Kenyon cells applies to both transgenes and will intrinsically limit their ability to induce reward-specific behavior, as aversive pathways might be triggered simultaneously.

In addition, we now show that optoDop1R1^{V1} cannot rescue rotenone-induced defects in larval locomotion, while Dop1R1^{V2} can virtually fully and specifically rescue behavior (see Fig. 5, S6). While it remains to be shown whether this is due to specificity, localization, or expression levels, we believe that the improved design will be beneficial for more sophisticated *in vivo* application and also future optoXRs design strategies.

In Figure 5, the authors show how activation of Dop1R1 boosts the activity during the first hours of the circadian cycle and indicate that "...we also tested optoDop1R2V2 activation under the same conditions but did not observe a significant effect on blue-light-induced activity (Fig. 5I, S5A-C)..." and conclude "These findings suggest a specific role for Dop1R1 signaling in I-LNVs promoting morning activity upon arousal." It would be interesting to test, using G protein-specific RNAi, whether the differential requirement is due to the levels of expression, cellular distribution, or G protein engaged by the optoDop1R2 construct. Is Dop1R1V1 also able to produce this effect on blue-light-induced activity?

Response: We appreciate the reviewer's suggestions regarding this point. While it will be interesting to test for the specific G proteins in I-LNV function in future studies, we think it is beyond the scope of our manuscript. Here, we aimed to provide an initial characterization of our tools in a broad set of assays. Similarly, while optoDop1R1^{V1} might be functional in this context, our goal was to generate improved tools, which we demonstrate in several assays showing improved function of the V2 variant in inducing cAMP signaling (Fig. 4, S5) and rescuing larval locomotion (Fig.5, S6).

Regarding the differential function of Dop1R1 vs Dop1R2 in I-LNvs neurons, a previous report from one of us has shown a specific function for Dop1R1 but not Dop1R2 in light-induced arousal⁹. Moreover, we have examined the expression of Dop1R1 and Dop1R2 in I-LNvs using Gal4 knock-in lines revealing the respective endogenous expression pattern (see Fig. S7f). Here, we found reporter expression for Dop1R1 specifically in I-LNvs, while Dop1R2 displayed only weak expression close to the reporter background level in this line. Based on the differential signaling properties and expression patterns of Dop1R1 and Dop1R2 we would thus expect differential functions, which are also reflected in our optoDopR results regarding light-induced arousal.

It would be helpful to explain why sometimes the activation of the optoDop receptors is performed with blue light and in other cases with green light.

Response: we thank the reviewer for this suggestion. While Rhodopsin and corresponding optoXRs chimera are best activated with blue light, short wavelength light exposure can cause unwanted behavioral effects due to its aversive/noxious nature of blue light in larvae^{10,11}, as well as lower penetration of the adult cuticle¹². We therefore preferentially used green light (525nm) to activate our optoDopRs, which has a much milder or no effect on behavior. However, in the case of PDF neuron-mediated light-induced arousal we used blue light for efficient induction of arousal as well as activation of optoDopRs. We now state this more clearly in our manuscript (see p18 l422ff).

Finally, in the discussion section, the authors claim that spectral differences between optoDop1R constructs and other light-induced proteins (like Crimson, for instance) would allow their combined activation and directly address the modulatory influence of dopamine on the activation parameters of specific neurons. This is fascinating, and it would be more attractive if the authors could propose a specific situation in which it could be implemented, such as epileptic conditions or learning processes, for example.

Response: we thank the reviewer for this suggestion, and we have added an example that could be tested regarding timing-dependent activation of dopaminergic neurons and responding DopRs to investigate synaptic and behavioral plasticity (p27 l610ff).

In line 353, a closing parenthesis is missing: "MBON-g1/g2 (adapted from 50." Reference 71 is incorrectly written.

Response: we have corrected these mistakes.

These are some critical points and concerns regarding the manuscript. It would be beneficial for the authors to address these issues and make the necessary revisions before considering for publication.

We thank all reviewers for their helpful and constructive feedback, and we believe we have fully and adequately addressed the raised concerns.

References

1. Aceves-Piña, E. O. & Quinn, W. G. Learning in Normal and Mutant *Drosophila* Larvae. *Science* (1979) **206**, 93–96 (1979).
2. Schroll, C. *et al.* Light-Induced Activation of Distinct Modulatory Neurons Triggers Appetitive or Aversive Learning in *Drosophila* Larvae. *Current Biology* **16**, 1741–1747 (2006).
3. Kondo, S. *et al.* Neurochemical Organization of the *Drosophila* Brain Visualized by Endogenously Tagged Neurotransmitter Receptors. *Cell Rep* **30**, 284–297.e5 (2020).
4. Ballister, E. R., Rodgers, J., Martial, F. & Lucas, R. J. A live cell assay of GPCR coupling allows identification of optogenetic tools for controlling Go and Gi signaling. *BMC Biol* **16**, 10 (2018).
5. Masuho, I. *et al.* Distinct profiles of functional discrimination among G proteins determine the actions of G protein-coupled receptors. *Sci Signal* **8**, 1–16 (2015).
6. Olsen, R. H. J. *et al.* TRUPATH, an open-source biosensor platform for interrogating the GPCR transducerome. *Nat Chem Biol* **16**, 841–849 (2020).
7. Yang, W., Hildebrandt, J. D. & Schaffer, J. E. G-proteins | Gq family. *Encyclopedia of Biological Chemistry: Third Edition* **6**, 450–455 (2021).
8. Handler, A. *et al.* Distinct Dopamine Receptor Pathways Underlie the Temporal Sensitivity of Associative Learning. *Cell* **178**, 60–75.e19 (2019).
9. Fernandez-Chiappe, F. *et al.* Dopamine Signaling in Wake-Promoting Clock Neurons Is Not Required for the Normal Regulation of Sleep in *Drosophila*. *The Journal of Neuroscience* **40**, 9617–9633 (2020).
10. Imambocus, B. N. *et al.* A neuropeptidergic circuit gates selective escape behavior of *Drosophila* larvae. *Current Biology* **32**, 149–163.e8 (2022).
11. Xiang, Y. *et al.* Light-avoidance-mediating photoreceptors tile the *Drosophila* larval body wall. *Nature* **468**, 921–6 (2010).
12. Inagaki, H. K. *et al.* Optogenetic control of *Drosophila* using a red-shifted channelrhodopsin reveals experience-dependent influences on courtship. *Nat Methods* **11**, 325–32 (2014).

REVIEWERS' COMMENTS

Reviewer #1 (Remarks to the Author):

The new version of the manuscript is satisfactory to me. Authors put in a lot of effort to improve text and figures. Just one comment - in the table 1 please add that Pdf-Gal4 strain has Gal4 expression in sLNv, too. I recommend this manuscript for publication in this form.

Reviewer #2 (Remarks to the Author):

I thank the authors for considering my comments and addressed them in their manuscript. I think that the manuscript significantly improved in clarity.

Reviewer #3 (Remarks to the Author):

Authors has covered all the questions and suggestions raised and therefore their manuscript has improved gratefully. Additionally, they have included new results regarding the use and functionality of the optoDop1R quimeras as well as a detailed description in the material and methods section.

All the mistakes of the previous version have been also corrected.

I strongly suggest this work should be accepted for publication in its present form.

Response to REVIEWERS' COMMENTS

We would like to thank all reviewers for their helpful comments allowing us to improve our work. Please find our detailed response below.

Reviewer #1 (Remarks to the Author):

The new version of the manuscript is satisfactory to me. Authors put in a lot of effort to improve text and figures. Just one comment - in the table 1 please add that Pdf-Gal4 strain has Gal4 expression in sLNv, too. I recommend this manuscript for publication in this form.

Response: We thank the reviewer for the helpful comments and very positive assessment of our work. We have added the statement of sLNv expression by Pdf-Gal4 in Table 1.

Reviewer #2 (Remarks to the Author):

I thank the authors for considering my comments and addressed them in their manuscript. I think that the manuscript significantly improved in clarity.

Response: We thank the reviewer for the helpful comments and very positive assessment of our work.

Reviewer #3 (Remarks to the Author):

Authors has covered all the questions and suggestions raised and therefore their manuscript has improved gratefully. Additionally, they have included new results regarding the use and functionality of the optoDop1R quimeras as well as a detailed description in the material and methods section.

All the mistakes of the previous version have been also corrected.

I strongly suggest this work should be accepted for publication in its present form.

Response: We thank the reviewer for the helpful comments and the strong endorsement of our work.